# Phenylalanine-tRNA aminoacylation is compromised by ALS/FTD-associated C9orf72 C4G2 repeat RNA

Mirjana Malnar Črnigoj [1,2,13], Urša Čerček[1,2,13], Xiaoke Yin [3], Manh Tin Ho[4], Barbka Repic Lampret[5], Manuela Neumann[6,7], Andreas Hermann [8,9], Guy Rouleau [10,11], Beat Suter[4], Manuel Mayr [3] & Boris Rogelj [1,12] ✉

The expanded hexanucleotide GGGGCC repeat mutation in the *C9orf72* gene is the main genetic cause of amyotrophic lateral sclerosis and frontotemporal dementia. Under one disease mechanism, sense and antisense transcripts of the repeat are predicted to bind various RNA-binding proteins, compromise their function and cause cytotoxicity. Here we identify phenylalanine-tRNA synthetase (FARS) subunit alpha (FARSA) as the main interactor of the CCCCGG antisense repeat RNA in cytosol. The aminoacylation of tRNA^Phe by FARS is inhibited by antisense RNA, leading to decreased levels of charged tRNA^Phe. Remarkably, this is associated with global reduction of phenylalanine incorporation in the proteome and decrease in expression of phenylalanine-rich proteins in cellular models and patient tissues. In conclusion, this study reveals functional inhibition of FARSA in the presence of antisense RNA repeats. Compromised aminoacylation of tRNA could lead to impairments in protein synthesis and further contribute to *C9orf72* mutation-associated pathology.

Multiple neurological disorders are caused by expansion of repeat sequences occurring in coding and non-coding DNA regions that can cause altered RNA or protein functions[1]. Mechanisms contributing to pathology include protein loss-of-function, and repeat sequence gain-of-function, such as production of toxic RNA and repeat associated non-AUG (RAN) translation[1]. Both loss and gain-of-function mechanisms have been proposed to contribute to the pathology of expanded repeat mutation in *C9orf72* gene[2]. Expanded hexanucleotide GGGGCC repeats in the first intron of *C9orf72* gene are the main genetic cause of the neurodegenerative diseases amyotrophic lateral sclerosis (ALS) and frontotemporal dementia (FTD) and are the most common cause of familial ALS and FTD within Europe and North America[3–6]. In healthy individuals 5–10 and up to 20–25 repeats occur, whereas in disease, they can reach several hundred to several thousand units[3,4,7]. However, the consensus on the exact threshold repeat number for disease development is still not reached. ALS cases with 24 or 28 repeats, and

[1]Department of Biotechnology, Jožef Stefan Institute, Ljubljana 1000, Slovenia. [2]Graduate School of Biomedicine, Faculty of Medicine, University of Ljubljana, Ljubljana 1000, Slovenia. [3]King's BHF Centre, King's College London, London SE5 9NU, UK. [4]Institute of Cell Biology, University of Bern, Bern 3012, Switzerland. [5]Clinical Institute of Special Laboratory Diagnostics, University Children's Hospital, University Medical Centre Ljubljana, Ljubljana 1000, Slovenia. [6]Molecular Neuropathology of Neurodegenerative Diseases, German Center for Neurodegenerative Diseases, Tübingen 72076, Germany. [7]Department of Neuropathology, University Hospital of Tübingen, Tübingen 72076, Germany. [8]Translational Neurodegeneration Section "Albrecht-Kossel", Department of Neurology and Center for Transdisciplinary Neurosciences Rostock (CTNR), University Medical Center Rostock, University of Rostock, 18147 Rostock, Germany. [9]Deutsches Zentrum für Neurodegenerative Erkrankungen (DZNE), Rostock/Greifswald, 18147 Rostock, Germany. [10]Department of Human Genetics, McGill University, Montréal, QC H3A 0G4, Canada. [11]Department of Neurology and Neurosurgery, Montreal Neurological Institute, McGill University, Montréal, QC H3A 0G4, Canada. [12]Faculty of Chemistry and Chemical Technology, University of Ljubljana, Ljubljana 1000, Slovenia. [13]These authors contributed equally: Mirjana Malnar Črnigoj, Urša Čerček. ✉e-mail: boris.rogelj@ijs.si

FTD cases of 20-22 repeats have been reported[3,4,7–9]. Expanded repeats are transcribed in both directions producing sense (G4C2) and antisense (C4G2) RNAs, which are predicted to sequester RNA-binding proteins, impede their normal function and trigger cytotoxicity[3,10–20]. Various proteins have already been identified to interact with these RNAs in transfected cells as well as in postmortem brain tissue[11,12,14,15,18,20–26]. However, previous studies focused mostly on the sense RNA repeats, although it is the antisense RNA that correlates better with TDP-43 pathology[22]. The interactions of repeat RNAs and proteins are mostly studied through co-localization of fluorescent signals from nuclear RNA foci and proteins in cells[11,12,14,15,18,20–26]. To our knowledge, co-localization with both sense and antisense RNA foci was shown for serine/arginine-rich splicing factor 1 and 2 (SRSF1, SRSF2), transcriptional activator protein Pur-alpha (PURA), THO complex subunit 4 (ALYREF), nucleolin (NCL), heterogeneous nuclear ribonucleoprotein H1/F (HNRNP H1/F), HNRNPA1, and HNRNPK[22,24,27]. The co-localizations of listed proteins were overall much lower for antisense foci than those for sense foci. Moreover, the antisense RNA constructs that were used for in vitro detection of direct interactions were short (4-5 repeats)[21,22,24]. Although most of the research so far was focused on nuclear interactors[11,12,14,15,18,20–26], these RNA repeats undergo RAN translation to toxic dipeptide repeat proteins (DPRs) and therefore, must also be present in the cytoplasm[10,23,28–30]. Additionally, repeats undergo length dependent nuclear export through binding to SRSF1 and NXF1[27].

Here we have identified protein interactors for disease-relevant length of antisense RNA (32 repeats) and assessed co-localizations throughout the cell with RNA-protein proximity ligation assay (PLA) detection in C9orf72 patient-derived cell lines. As protein synthesis is often compromised in expanded repeat disorders[1] and aminoacyl-tRNA synthetases (ARSs) are increasingly implicated in nervous system disorders including Charcot-Marie-Tooth disease, cerebellar ataxia and possibly Alzheimer's disease[31,32], we have studied the interaction of antisense RNA with phenylalanine-tRNA (tRNA$^{Phe}$) synthetase (FARS) in more detail. Cytoplasmic FARS is the only ARS that is a hetero-tetramere, composed of a catalytic subunit alpha (FARSA) and editing domain beta (FARSB)[33]. Mutations in either subunit of the FARS protein are rare due to high lethality but show a neurodegenerative phenotype[34,35]. In Drosophila they caused protein mistranslation, ER stress, and neuronal loss[36]. With this study we focus on the catalytic FARSA subunit which executes tRNA-aminoacylation as it shows greater propensity for binding C4G2 repeats than editing FARSB subunit. Aminoacylation assays performed in this study on isolated protein in the presence of antisense RNA and in C9orf72 patient-derived cells indicate inhibition of FARSA tRNA-aminoacylation function. Remarkably this is associated with overall reduction of phenylalanine (Phe) abundance in the total proteome of antisense repeat over-expressing cells and decreased expression of proteins with high Phe content in FARSA knockdown cells, C9orf72 patient-derived cell lines, and post-mortem cerebellum tissue from C9orf72 ALS and FTD patients.

## Results

### Antisense repeat RNA interacts with FARS
We have determined the interactors of antisense RNA repeats with RNA pull-down assays in combination with mass spectrometry (MS). The following in vitro transcribed RNA constructs were used: 32×C4G2 repeats with the S1m aptamer on the 3′-end (32×C4G2-S1m) and the S1m aptamer control (S1m). A schematic representation of the RNA constructs and RNA pull-down assay is presented in Fig. 1a. We verified the integrity and purity of RNA constructs with agarose gel electrophoresis (Supplementary Fig. 1a). RNA pull-down assays were performed on cytoplasmic and nuclear fractions of mouse brain lysate (Supplementary Fig. 1b). Eluates were separated with SDS-PAGE and the gels were silver stained (Fig. 1b). Distinct bands were cut out

(labelled on the gels in Fig. 1b) and analyzed with MS (Supplementary Table 1). The protein candidates for further analysis were selected according to the cut-off criteria of a spectral count of >20 and fold-change of >3 (Supplementary Table 1).

Following MS identification of interacting proteins, we performed RNA pull-down assays on protein lysates from human postmortem brain tissue to confirm the interactions. An additional RNA construct with a random control sequence of equivalent length to the 32×C4G2-S1m RNA was used for this purpose—the partial sequence of red fluorescent protein with the S1m aptamer on the 3′-end (RFP-S1m). Using western blots, we verified selected proteins in eluates from RNA pull-down assays. Interactions of C4G2 RNA construct with the following proteins were detected in multiple assays: serine/threonine-protein kinase TAO1 (TAOK1), cytoplasmic FMR1-interacting protein 1/2 (CYFIP1/2), heterogeneous nuclear ribonucleoprotein L (HNRNPL), 2′,3′-cyclic nucleotide 3′-phosphodiesterase (CNP), nucleophosmin 1 (NPM1), FARS subunits alpha (FARSA) and beta (FARSB) (Fig. 1b, c, Supplementary Fig. 1c). We could not validate the interaction between antisense repeat RNA and seryl-tRNA synthetase (SARS) in the western blot analysis and this was not investigated further (Supplementary Fig. 1c).

Following in vitro assays, interactions of the antisense RNA and selected proteins were analyzed in C9orf72 patient-derived and control cell lines. We developed a modified RNA-protein PLA to observe the localization of chosen proteins relative to endogenous antisense RNA in C9orf72 patient-derived cells to detect both nuclear and cytoplasmic interactions (Fig. 2a). C9orf72 patient-derived and control fibroblasts, lymphoblastoid cells, and iPSCs were used for this assay (Fig. 2b–d, Supplementary Fig. 2, Supplementary Tables 2–4). We analyzed the interactions of antisense repeat RNA with proteins identified in this study (FARSA, FARSB, TAOK1, HNRNPL) and with proteins found in previous studies (HNRNPK)[22,24]. RNA-protein PLA signal was increased in all analyzed C9orf72 patient-derived cells relative to control for antisense RNA-FARSA interaction. Namely, the fold change increase relative to control was as follows: $3.53 \pm 0.36$ in fibroblasts (Fig. 2b), $10.98 \pm 1.01$ in lymphoblastoid cells (Fig. 2c), and $27.95 \pm 6.63$ in iPSCs (Fig. 2d). In addition, there were increased PLA signals in C9orf72 patient-derived fibroblasts compared to control for TAOK1 ($3.09 \pm 0.06$), HNRNPL ($4.66 \pm 0.41$), and the previously identified antisense RNA interactor HNRNPK ($1.47 \pm 0.10$) (Supplementary Fig. 2a). Interestingly, we have not detected an increase in PLA signal for FARSB (Supplementary Fig. 2a), which suggests indirect interaction of FARSB with antisense repeats via FARSA as FARSB was among proteins detected in RNA pull-down assay (Fig. 1b, c, Supplementary Fig. 1c).

To determine the specificity of interaction with FARSA, we also assessed interactions with other ARSs, with RNA-protein PLA. There was no increase in signal for the interaction of endogenous antisense RNA and asparagine-tRNA synthetase (NARS), leucine-tRNA synthetase (LARS), and glutamyl-prolyl-tRNA synthetase (EPRS) in C9orf72 patient-derived and control fibroblasts (Supplementary Fig. 2b), lymphoblastoid cells (Supplementary Fig. 2c), and iPSCs (Supplementary Fig. 2d).

To further evaluate FARSA interaction with antisense RNA, we performed classic RNA-FISH analysis in combination with IF in C9orf72 patient-derived and control fibroblasts. Fibroblasts were chosen, as we detected also nuclear FARSA in these cells. Co-localization was evaluated by fluorescence signal overlap. In accordance with the RNA-protein PLA, there was significant co-localization of FARSA with nuclear antisense repeat RNA foci (Supplementary Fig. 3).

To substantiate the interaction between FARSA and antisense RNA, we performed RNA immunoprecipitation (RIP) assay on C9orf72 patient-derived fibroblasts using FARSA antibody. A fold change of $4.90 \pm 0.32$ was observed for antisense RNA present in the FARSA RIP compared to control rabbit IgG RIP (Supplementary Fig. 1d).

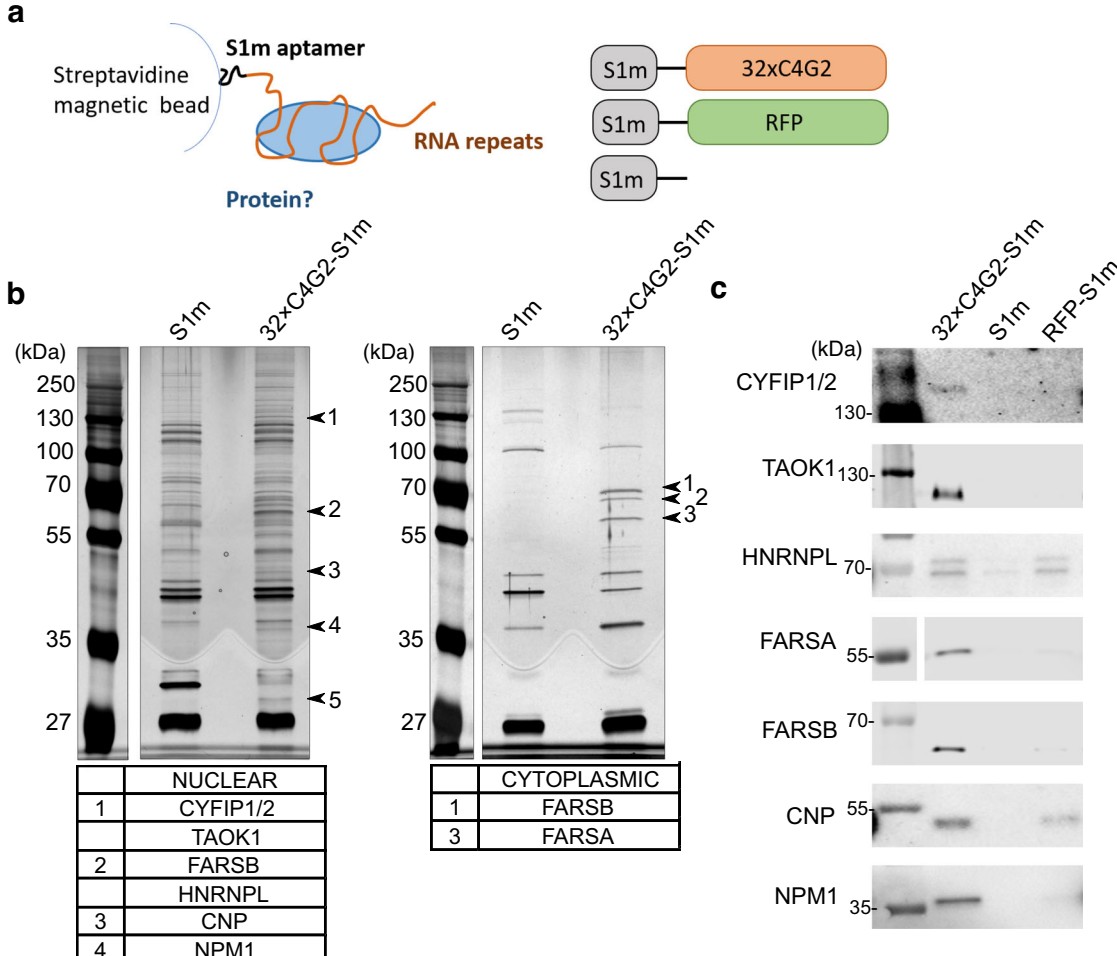

**Fig. 1 | Antisense RNA from the C9orf72 mutation binds various proteins.**
**a** Schematic representation of the RNA pull-down assay and RNA constructs used in the experiments. **b** Silver stained SDS-PAGE gels with labelled bands analyzed with mass spectrometry. The proteins were obtained with RNA pull-down assay performed on the nuclear and cytoplasmic fractions of mouse brain lysate. The RNA constructs 32×C4G2-S1m and S1m were used. Results were repeated in two independent experiments. **c** The identified proteins detected as interactors with western blot from the RNA pull-down assay performed on protein lysates from human postmortem brain tissue with 32×C4G2-S1m and control RFP-S1m and S1m RNA constructs. Results were repeated in two independent experiments. Source data are provided as a Source data file.

To determine whether different length of the repeats affects the binding of repeat RNA and FARSA we performed RNA-pull down assay from HEK293 cell lysate using different length of C4G2 repeats (Supplementary Fig. 1e). We observed the strongest signal for FARSA binding 32x C4G2 repeat RNA. The binding to 24× repeats was present, but to lower extent, whereas the binding to 8× C4G2 was not increased compared to control RFP construct.

**Aminoacylation of tRNA$^{Phe}$ is decreased in the presence of C4G2 RNA repeats**

Potential functional impact of antisense repeat RNA on FARS catalytic function was assessed using two different aminoacylation assays. First, in vitro tRNA aminoacylation assays were performed using recombinant FARSA and FARSB proteins. The presence of 32×C4G2 RNA repeats in three different concentrations (0.04 ng/ul, 0.4 ng/μL and 4 ng/μL) inhibited the aminoacylation of tRNA$^{Phe}$ in this system (using 43.75 ng/μL FARS protein), which was not the case for the control RFP RNA used at the same concentration (Fig. 2e). Aminoacylation of tRNA$^{Phe}$ was assessed at multiple time points from 10 to 90 min. The reaction reached plateau between 60 and 90 min, at which point it was still lower in the presence of 32×C4G2 RNA repeats than in controls.

In the second assay we assessed the aminoacylation function of FARS in C9orf72 patient-derived cells. We determined the percentage of charged tRNA$^{Phe}$ in extracts from C9orf72 patient-derived and control lymphoblastoid cells according to the protocol of Loayza-Puch et al.[37]. Significantly decreased levels of Phe-tRNA$^{Phe}$ (charged tRNA$^{Phe}$) were detected in C9orf72 patient-derived cells as $0.77 \pm 0.03$ tRNA$^{Phe}$ was charged relative to control cells (Fig. 2f). Notably, expression of FARSA did not significantly differ between C9orf72 patient-derived and control lymphoblastoid cells (Supplementary Fig. 4b). In addition, we tested the tRNA-aminoacylation levels for three other tRNAs, namely tRNA$^{Asn}$, tRNA$^{Leu}$, and tRNA$^{Pro}$, and observed no significant changes in any of these. Compared to controls, the charging levels of these tRNAs were as follows: $0.93 \pm 0.17$ (tRNA$^{Asn}$), $0.78 \pm 0.11$ (tRNA$^{Leu}$), and $0.84 \pm 0.32$ (tRNA$^{Pro}$) (Fig. 2f). All aminoacylation levels were normalized to specific amino acid levels in C9orf72 patient-derived and control cells, as to eliminate the amino acid shortage as cause of differences in aminoacylation (Fig. 2g, Supplementary Fig. 4a). This observation is in accordance with the results from RNA-protein PLA, as NARS, LARS, and EPRS were not found to interact with antisense RNA in C9orf72 patient-derived cells (Supplementary Fig. 2b–d). Moreover, we tested binding of NARS, LARS, and EPRS to 32×C4G2 RNA with RNA pull-down assay, and no binding was observed in any of these cases (Supplementary Fig. 5).

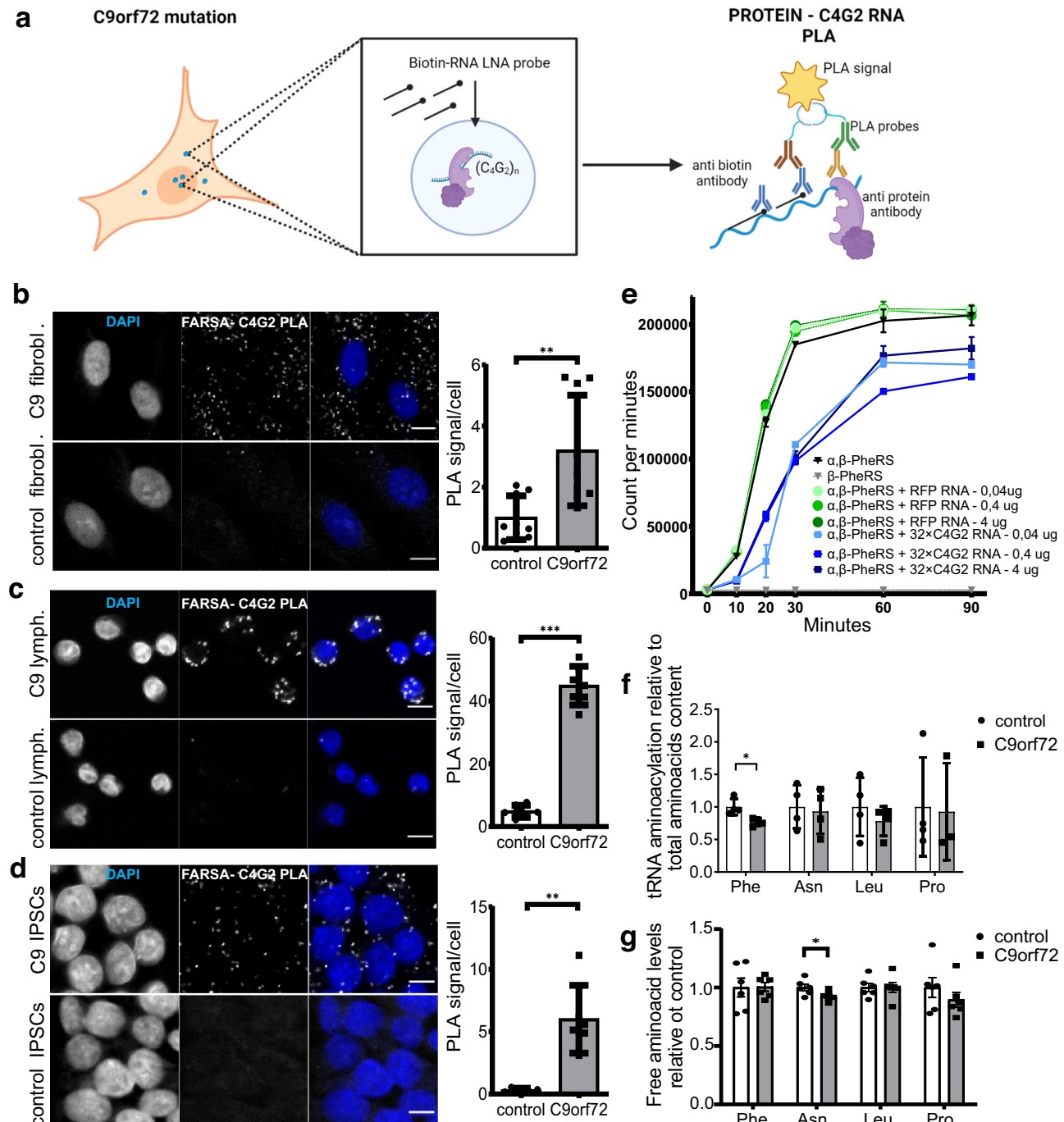

## Antisense RNA negatively impacts phenylalanine content of the proteome

The observed inhibitory impact of C4G2 RNA repeats on the function of recombinant FARS protein (Fig. 2e) and the decreased Phe-tRNA$^{Phe}$ levels in C9orf72 patient-derived cells (Fig. 2f) indicated potential consequences of this interaction on protein synthesis. Therefore, we evaluated changes in Phe incorporation at the proteome level using a mutated FARSA which incorporates azido-Phe into newly synthetized proteins[38]. Using click chemistry, we could detect the levels of incorporated azido-Phe. Firstly, we showed that mutated FARSA binds 32×C4G2 repeat RNA and not control RNAs in RNA-pull down experiments in a HEK293T cell line transfected with the pcDNA3.1-mFARSA-HA construct (Supplementary Fig. 6a). Click chemistry experiments showed around 20% reduction in expression of Phe levels when the stable cell line HEK293T – mFARSA was transfected with 32×C4G2 repeat RNA

compared to RFP (Fig. 3a). Experiment was repeated in four independent repeats. Results for expression levels normalized to whole protein expression and relative to S1m transfected cells were for RFP 105.1 ± 6.83 and for 32×C4G2 81.4 ± 5.64 (Fig. 3a, Supplementary Fig. 6b).

Additionally, we tested the expression levels of selected proteins with the highest content of Phe (3σ over average % Phe in the proteome) (from now on Phe-rich proteins) in different cell lines and post-mortem tissue from C9orf72-mutation carriers. Proteins were chosen based on the analysis of their amino acid content[39], expected expression levels gauged from the Human Protein Atlas (https://www.proteinatlas.org/), and commercial availability of antibodies. We analyzed the following Phe-rich proteins: TSPAN5 (10.1% Phe), PXMP2 (10.3% Phe), GOLT1B (17.4% Phe), and ALG10B (12.1% Phe) and Phe-low proteins: LAMINB1 (0.7%), and GAPDH (4.2%) (percentage of Phe to the total amino acid content of the respective protein is given in brackets).

**Fig. 2 | FARSA interacts with antisense RNA in the nucleus and cytoplasm of C9orf72 patient-derived cells and the presence of C4G2 RNA reduces aminoacylation of tRNA$^{Phe}$. a** Schematic representation (created with Biorender.com) of RNA-protein proximity ligation assay (PLA). Increased PLA signals in C9orf72 patient-derived cells with expanded repeats relative to controls: $0.78 \pm 0.36$ in three control ($n_{cells} = 545$) versus $2.49 \pm 0.92$ in three C9orf72 patient-derived ($n_{cells} = 600$) fibroblasts cell lines in 3 independent experiments (unpaired Student's $t$-test, $p$-value (two-sided) = 0.006) (**b**), $4.85 \pm 0.77$ in three control ($n_{cells} = 1010$) versus $44.98 \pm 3.95$ in three C9orf72 patient-derived ($n_{cells} = 986$) lymphoblastoid cell lines in 3 independent experiments (unpaired Student's $t$-test, $p$-value (two-sided) = 0.0000006) (**c**), and $0.31 \pm 0.08$ in one control ($n_{cells} = 3818$) versus $6.00 \pm 1.12$ in one C9orf72 patient-derived ($n_{cells} = 2863$) induced pluripotent stem cells (iPSCs) in 3 independent experiments (unpaired Student's $t$-test, $p$-value (two-sided) = 0.004) (**d**). For all three cell types, C9orf72 patient-derived and control cells are displayed above and below, respectively. **e** In vitro aminoacylation assays performed with recombinant FARS protein (43.75 ng/μL) showed decreased activity of tRNA$^{Phe}$ charging with Phe in the presence of 32×C4G2 RNA repeats (blue squares) as compared to control RFP RNA (green circles) at three different concentrations of RNA: 0.04 ng/μL, 0.4 ng/μL and 4 ng/μL (shades of blue and green). Positive control reaction containing both FARSA and FARSB subunits but without any RNA constructs is presented with black lines and triangles. Negative control with only FARSB is presented with gray lines and triangles. The experiment was repeated 3 times. **f** Discriminating between aminoacylated and free tRNA 3' ends revealed a significant decrease in tRNA$^{Phe}$ aminoacylation in C9orf72 patient-derived versus control lymphoblastoid cell lines, relative to total aminoacids content: $0.77 \pm 0.03$ (Phe-tRNA$^{Phe}$) ($p$-value (two-sided) = 0.026), $0.93 \pm 0.17$ (Asn-tRNA$^{Asn}$) ($p$-value (two-sided) = 0.782), $0.78 \pm 0.11$ (Leu-tRNA$^{Leu}$) ($p$-value (two-sided) = 0.429), and $0.84 \pm 0.32$ (Pro-tRNA$^{Pro}$) ($p$-value (two-sided) = 0.761). Statistics was done using unpaired Student's $t$-test. The concentration of respective amino acid in C9orf72 patient-derived and control lymphoblastoid cell lines was used for normalization. Experiment was performed on four C9orf72 patient-derived and four control biologically independent cell lines. **g** The concentration of free Phe did not significantly differ between C9orf72 patient-derived and control lymphoblastoid cells. Concentration of free aminoacids relative to total aminoacid levels: Phe ($1.00 \pm 0.04$; $p$-value (two-sided) = 0.957), Asn ($0.92 \pm 0.02$; $p$-value (two-sided) = 0.046), Leu ($1.00 \pm 0.04$; $p$-value (two-sided) = 0.99), Pro ($0.90 \pm 0,06$; $p$-value (two-sided) = 0.352). Experiment was performed on six C9orf72 patient-derived and six control cell lines. Scale bars: 10 μm. Graphs present mean values ± s.e.m. with statistical significance labeled as: *$p < 0.05$, **$p < 0.01$, ***$p < 0.001$. Source data are provided as a Source data file.

In order to evaluate if partial loss of FARSA function results in lower expression of Phe-rich proteins, we quantified their expression levels in FARSA knockdown HEK293 cells (Fig. 3b, Supplementary Fig. 7). In a partial knockdown with $50.3 \pm 0.5\%$ FARSA expression (Supplementary Fig. 7a) we observed a significant reduction in expression levels of selected Phe-rich proteins relative to average shScramble treated cells (100%): TSPAN5 ($76.2 \pm 10.6\%$), PXMP2 ($81.1 \pm 10.9\%$), GOLT1B ($87.5 \pm 3.1\%$), ALG10B ($77.9 \pm 9.4\%$) but not controls: GAPDH ($101.6 \pm 6.5\%$) and LAMINB1 ($106.1 \pm 7.1\%$) (Fig. 3b, Supplementary Fig. 7b). Four independent knockdown experiments with three technical replicates were performed and loading comparison was done based on whole protein levels from stain free membrane.

To determine if the presence of C4G2 RNA affects cells equivalently to partial loss of function we further evaluated the reduction of Phe-rich proteins in C9orf72-patient derived cells. We performed western blot analysis on 6 C9orf72-mutation positive and 6 control lymphoblastoid cell lines. We observed similar effect of reduction in Phe-rich protein expression (Fig. 3c). Expression levels of Phe-rich proteins were lower in C9orf72 patient-derived cells compared to controls as follows: $14.41 \pm 4.46$ in control and $8.63 \pm 3.55$ in C9orf72 patient-derived cells for TSPAN5; $10.1 \pm 2.15$ in control and $3.88 \pm 1.68$ in C9orf72 patient-derived cells for PXMP2; $15.75 \pm 3.6$ in control and $11.22 \pm 1.88$ in C9orf72 patient-derived cells for GOLT1B; $9.39 \pm 0.27$ in control and $8.44 \pm 0.24$ in C9orf72 patient-derived cells for ALG10B; $9.350 \pm 0.508$ in control and $10.036 \pm 0.507$ in C9orf72 patient-derived cells for LAMINB1; $15.167 \pm 0.586$ in control and $14.430 \pm 0.714$ in C9orf72 patient-derived cells for GAPDH. The experiment was performed in two technical repeats (Fig. 3c, Supplementary Fig. 8a) and loading comparison was done based on whole protein level from stain free membrane. Statistical significance was calculated using unpaired Student's $t$-test between different cell lines and with mixed models' analysis between two technical repeats showing a significant reduction in expression for all Phe-rich proteins. qPCR analysis of mRNA transcripts revealed that the change in expression was not due to lower levels of transcripts, since we have not detected any significant difference between C9orf72 patient-derived lymphoblastoid cell lines and controls (Supplementary Fig. 8b).

Additionally, we knocked down FARSA in 7-days differentiated NSC-34 cells, a differentiated motor neuron-like cell model (Fig. 4a, Supplementary Fig. 9). In a partial knockdown with $74 \pm 5.9\%$ FARSA expression level (Supplementary Fig. 9a) we observed reduction in expression of all tested Phe-rich proteins. Apart from TSPAN5, tested proteins showed significant decrease in expression in three biological replicates. Levels relative to average shScramble treated cells (100%)

were as follows: TSPAN5 ($84.2 \pm 9.8\%$), PXMP2 ($83.5 \pm 5.0\%$), GOLT1B ($70.0 \pm 8.5\%$), ALG10B ($74.4 \pm 1.3\%$), GAPDH ($109.6 \pm 11.3\%$) and LAMINB1 ($125.3 \pm 21.2\%$). For PXMP2, distinct band at around 10 kDa higher molecular weight as the one observed in human cell lines, was analyzed since there was no other band detected. Three independent knockdown experiments with three technical replicates were performed and loading comparison was done based on whole protein level from stain free membrane.

To link the FARSA dysfunction to C9orf72 associated ALS/FTD we evaluated the expression levels of Phe-rich proteins in lysates from post-mortem cerebellum from 12 ALS/FTD cases with presence of *C9orf72* mutation and 12 TDP-43-positive ALS/FTD cases without a *C9orf72* mutation (Fig. 4b, Supplementary Fig. 8c). Expression levels relative to average TDP-43-positive but *C9orf72* mutation-negative ALS/FTD were significantly reduced for two of the observed Phe-rich proteins: ALG10B ($53.8 \pm 10.8\%$) and TSPAN5 ($60.3 \pm 5.1\%$). Expression level was also significantly reduced for Phe-low protein GAPDH ($76.7 \pm 4.0\%$) but not LAMINB1 ($94.1 \pm 9.6\%$). Loading comparison was done based on whole protein level from stain free membrane. The expression of GOLT1B was not quantified as the antibody did not work well on human cerebellum samples (Supplementary Fig. 8c). Antibody for PXMP2 detected a band at the expected MW but also showed increased unspecific staining in upper MW (Source data file). We quantified the band at expected MW and observed non-significant reduction. All experiments investigating Phe-rich and Phe-low protein expression are summarized in Table 1.

In line with recently published papers showing that mutation in other ARS (GARS) related to neurodegenerative phenotype is connected to activation of stress response[40,41], we looked into this possible mechanism. qPCR analysis of genes *ASNS*, *GPT2* and *eif4EBP1* transcribed in stress response[42], showed no increased transcription in C9orf72 patient derived lymphoblastoid cells and in FARSA knockdown HEK293 and differentiated NSC-34 cells (Supplementary Fig. 10a–c). Additionally, we did not detect the formation of stress granules or mislocalisation of TDP-43 in FARSA knockdown HEK293 cells (Supplementary Fig. 10d).

## Number of downregulated genes increases with % Phe in proteins

In order to observe the change in expression of Phe rich proteins on whole proteome level we performed MS analysis of RIPA lysates from six C9orf72- patient derived and six control lymphoblastoid cell lines. We did not detect significant change (corrected $p$-value below 0.05) in expression level of proteins (Supplementary Data 1). Out of the four investigated Phe-rich proteins only GOLT1B and PXMP2 were detected

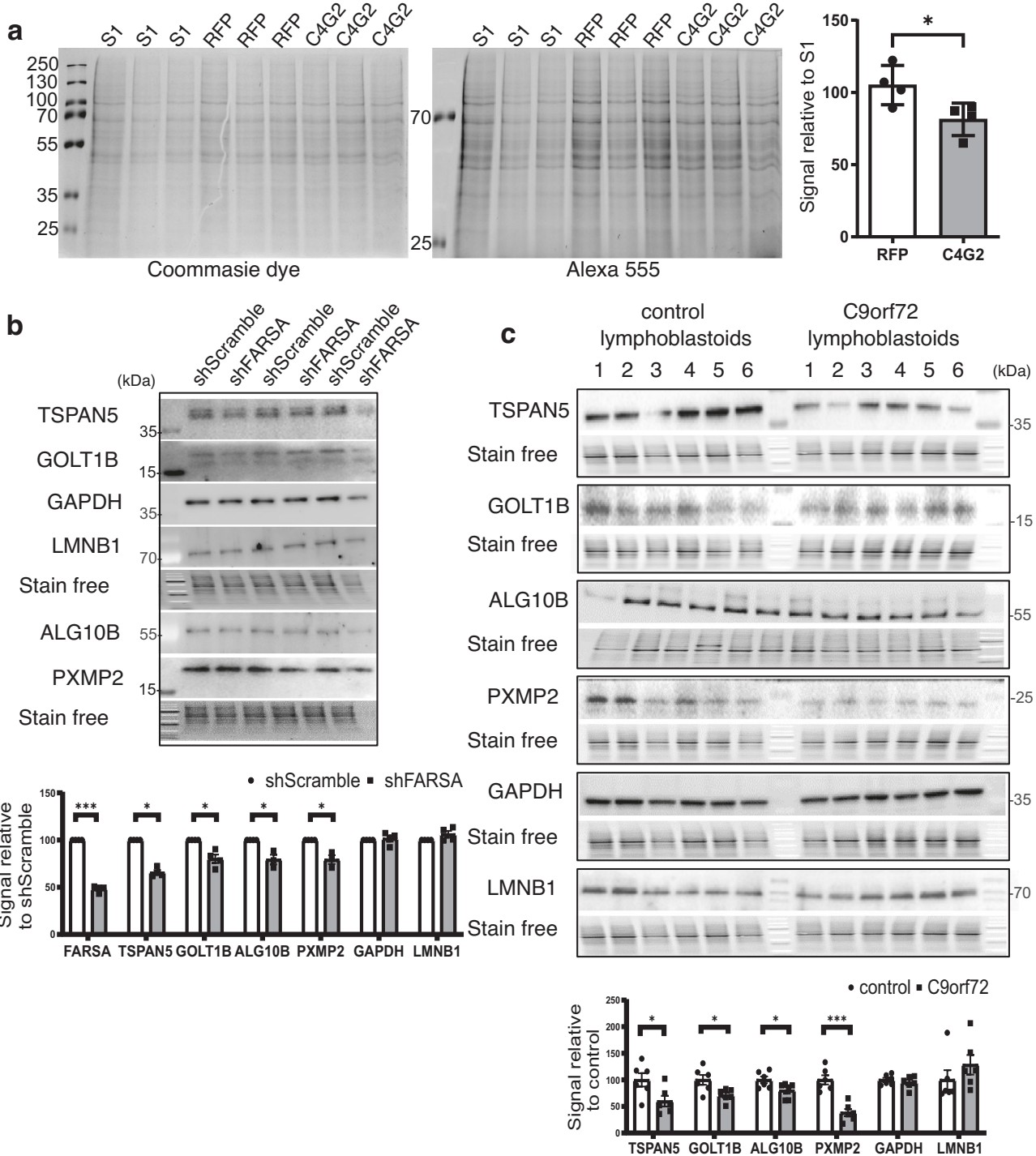

in the analysis. There is a trend showing the downregulation of these genes in C9orf72-patient derived lymphoblastoid cell lines but the change is non-significant. Since there were no significant changes we investigated if the increasing % of Phe in proteins correlates with the increasing number of downregulated genes in C9orf72-patient derived lymphoblastoid cells. We therefore selected protein candidates according to the cut-off criteria log2 fold-change of at least 0.15 which correlates to around 10% change in expression level. With this criteria also Phe-rich proteins were included in the analysis. We determined the % of Phe in each protein based on the data from[39] and calculated the % of downregulated and upregulated proteins in increasing 2% Phe intervals (Fig. 4c). We observed an increase in ratio of downregulated compared to upregulated proteins with Phe content over 6%.

## Discussion

In this study, we identified FARSA as the major cytoplasmic interactor of ALS/FTD-associated C9orf72 antisense RNA. The interaction leads to a reduction in tRNA$^{Phe}$ charging, which leads to decreased Phe incorporation into proteins in cell models and human post-mortem tissue of *C9orf72* mutation positive ALS/FTD patients. Presented proof of compromised protein synthesis could potentially have important implications for the development of *C9orf72* mutation related diseases. We showed that C4G2 repeats pull down both subunits of the FARS protein, but a direct interaction was confirmed only for FARSA in the subsequent analysis, suggesting that FARSB subunits interact indirectly with antisense RNA through their complex with FARSA subunits. Furthermore, our analysis shows that interaction of FARSA

**Fig. 3 | Expression of Phe-containing proteins is decreased in HEK293T cells, FARSA knockdown HEK293 cells and C9orf72 patient-derived lymphoblastoid cell lines. a** Click chemistry experiment in HEK293T cell line stably expressing mutated FARSA protein showed reduced global expression levels of Phe-containing proteins in the presence of 32×C4G2 RNA. Phe-content normalized to whole protein expression and relative to S1m transfected cells was for RFP 105.1 ± 6.83 and for 32×C4G2 81.4 ± 5.64 (*p*-value (two-sided) = 0.037). Statistics was done using unpaired Student's t-test on four independent experiments. The Western blot analysis of Phe-rich protein expression levels in (**b**) FARSA knockdown HEK293 cells and (**c**) C9orf72 patient-derived lymphoblastoid cells showed decreased expression of Phe-rich proteins TSPAN5, PXMP2, GOLT1B, and ALG10B whereas there was no decrease in expression of Phe-low proteins LAMINB1 and GAPDH. Expression levels in FARSA knockdown HEK293 cells relative to shScramble average (100%) are as follows: TSPAN5 (76.2 ± 10.6%; *p*-value = 0.023), PXMP2 (81.1 ± 10.9%; *p*-value = 0.026), GOLT1B (87.5 ± 3.1%; *p*-value = 0.048), ALG10B (77.9 ± 9.4%; *p*-value = 0.049), GAPDH (101.6 ± 6.5%; *p*-value = 0.897) and LAMINB1 (106.1 ± 7.1%; *p*-value = 0.249). Statistics was done using nested *t*-test on 4 independent

experiments. Expression levels in lymphoblastoid cell lines are as follows: 14.41 ± 4.46 in control and 8.63 ± 3.55 in C9orf72 patient-derived cells for TSPAN5 (*p*-value (two-sided) = 0.033); 10.1 ± 2.15 in control and 3.88 ± 1.68 in C9orf72 patient-derived cells for PXMP2 (*p*-value (two-sided) = 0.00028); 15.75 ± 3.6 in control and 11.22 ± 1.88 in C9orf72 patient-derived cells for GOLT1B (*p*-value (two-sided) = 0.0027); 9.39 ± 0.27 in control and 8.44 ± 0.24 in C9orf72 patient-derived cells for ALG10B (*p*-value (two-sided) = 0.038); 9.350 ± 0.508 in control and 10.036 ± 0.507 in C9orf72 patient-derived cells for LAMINB1 (*p*-value (two-sided) = 0.362); 15.167 ± 0.586 in control and 14.430 ± 0.714 in C9orf72 patient-derived cells for GAPDH (p-value (two-sided) = 0.444). Statistics were done using unpaired Student's *t*-test on six C9orf72 patient-derived and six control biologically independent samples. Stain free images represent loading comparison for all proteins above the stain free image and are vertically compressed for design reasons. Molecular weight is marked on western blot pictures in kDa. Graphs present mean values relative to shScramble in HEK293 or control lymphoblastoid cell lines ± s.e.m. with statistical significance labeled as *$p < 0.05$, **$p < 0.01$, ***$p < 0.001$. Source data are provided as a Source data file.

with antisense RNA is length dependent, further pointing to disease relevance of this interaction. To date, studies of C9orf72 repeat RNAs have focused predominantly on the sense repeats[3,4,7,10,13–18,23,24,28,43] although the antisense repeat RNA from this mutation has been shown to correlate more strongly with TDP-43 pathology[22]. Importantly, these studies have used FISH combined with immunofluorescence as the main method of validation of the RNA pulldown experiments, biasing the results towards validated nuclear interactions as the C9orf72 foci are predominantly nuclear. However, the presence of RNAs with extended repeats in the cytoplasm, where they undergo RAN translation[10,23,28–30], raises the question of what the interactors of the repeats are outside the nucleus. To overcome this issue, we have modified the RNA-protein proximity ligation assay and validated also the cytosolic interactions of the C4G2 repeat RNA with the identified interactors including FARS. Using in vitro FARS tRNA aminoacylation assay we showed that the presence of C4G2 RNA repeats can inhibit the charging of tRNA^Phe by recombinant FARS protein. The disease relevance of this inhibition was shown with C4G2 RNA-expressing C9orf72 patient-derived cells, where we observed reduced levels of charged tRNA^Phe compared to control cells. The level of free Phe available in cells was not raised as it is most likely compensated by the cell homeostasis mechanisms[44].

Expression of C4G2 RNA repeats in HEK293T cells resulted in decreased incorporation of Phe at the whole proteome level. Furthermore, we confirmed this effect in C9orf72 patient-derived cells and human post-mortem cerebellum tissue from C9orf72 mutation positive ALS/ FTD patients showing reduced expression levels of proteins with high Phe content. Similar results were obtained in loss of function FARSA knockdown models, including motor neuron-like cell model. On a side note, in cerebellum tissue we observed a significant decrease in expression of Phe-low protein GAPDH, whose dysfunctions has been observed before[45,46]. Interestingly, proteomic analysis in C9orf72-patient derived cell lines indicates that the percentage of downregulated genes increases with the percentage of Phe in proteins.

It remains to be determined whether any of the tested or potentially other Phe-rich proteins are individually causatively associated with ALS/FTD. However, the observed reduction on Phe incorporation may also present additional pressure to ageing related changes in proteostasis mechanisms implicated in the disease[47]. Each of the observed proteins have functions that could be implicated in neurodegeneration. TSPAN5 belongs to a family of tetraspanin proteins involved in the localization of disintegrin and metalloprotease 10 (ADAM10). It regulates the activity and expression of NOTCH proteins which are differentially expressed in ALS or ALS-FTD and the C9orf72 mutation is proposed to be one of the causing factors[48–50]. ALG10B belongs to a family of glucosyltransferase involved in N-glycosylation. Changes in N-glycosylation pattern of TAU and NPTX, associated with

the development of AD, FTD and PD, have been previously shown[51,52]. PXMP2 is a peroxisome channel-forming protein. Peroxisomes are involved in fatty acid oxidation, lipid synthesis, ROS metabolism and oxidative response and their dysfunction was observed in patients with AD[53,54]. GOLT1B is a protein involved in Golgi transport. Disruptions in Golgi function have already been implicated in connection to several neurodegenerative diseases including ALS[55,56]. Although previous studies show that increased levels of uncharged tRNAs and mutations in other ARSs, related to neurodegenerative phenotype lead to activation of stress response[40–42,57] we did not observe such response. This suggests on either a subtler mechanism that adds on to ageing related proteostatic changes or other pathways. In addition to the canonical function of FARSA (tRNA aminoacylation), other non-canonical functions of ARS may also be affected by this interaction including RNA processing and trafficking, apoptosis, inflammation, transcriptional regulation, cell signaling, autoimmune response, and proliferation[58–61].

In conclusion, the results of this study suggest that C4G2 repeat RNA may have important consequences for protein synthesis in C9orf72 mutation carriers (Fig. 4d). Further studies of the uncovered reduction of FARS activity are needed to determine the potential impact on protein synthesis and aggregation of proteins in ALS /FTD and to elucidate the full role of FARS and its interaction with the expanded RNA repeats of the *C9orf72* gene mutation in disease pathology. Additionally, functional association of neurodegenerative phenotype with here investigated Phe-rich proteins, each of which has potential disease-related functions, opens promising avenues for future research on *C9orf72* mutation associated pathology.

## Methods
### Reagents
The following primary antibodies and dilutions were used: FARSA (18121-1-AP; WB 1:1000; IF/PLA 1:500), FARSB (16341-1-AP; WB 1:1000; IF/PLA 1:500), TAOK1 (23379-1-AP; WB 1:1000, IF/PLA 1:500), CNP (13427-1-AP; WB 1:1000), CYFIP1/2 (16011-1-AP; WB 1:1000), GAPDH (60004-1-Ig, lot: 10013030 and 10494-1-AP; WB 1:5000), TSPAN5 (12122-1-AP; WB 1:500), PXMP2 (24801-1-AP; WB 1:600), LARS (21146-1-AP; WB 1:500, PLA 1:100), NARS (14882-1-AP; WB 1:500, PLA 1:100), EPRS (25307-1-AP; WB 1:500, PLA 1:100), HA (66006-2-ig, lot: 10011878; WB, 1:1000) (all from Proteintech, Rosemont, Illinois, USA); HNRNPL (sc-32317, lot: 4D11; WB 1:500, IF 1:100), HNRNPK (sc-28380, lot: D-6; WB 1:500, IF 1:100), NPM1 (sc-56622; WB 1:100), Lamin B1 (sc20682, lot: H90; WB 1:5000), Syp/Synaptophysin (sc-17750, lot: D0419; WB 1:1000) (all from Santa Cruz Biotechnology, Dallas, Texas, USA); Histone H3 (NB500-171; WB 1:1000), ALG10B (NBP3-09560; WB 1:500) (all from Novus Biologicals, Centennial, Colorado, USA; Anti-Biotin antibody [Hyb-8] (ab201341, lot: GR3446252-1; PLA 1:100), TDP-43 (ab80608; lot: GR83948-15; ICC 1:250) (all from Abcam, Cambridge, UK); Anti-biotin

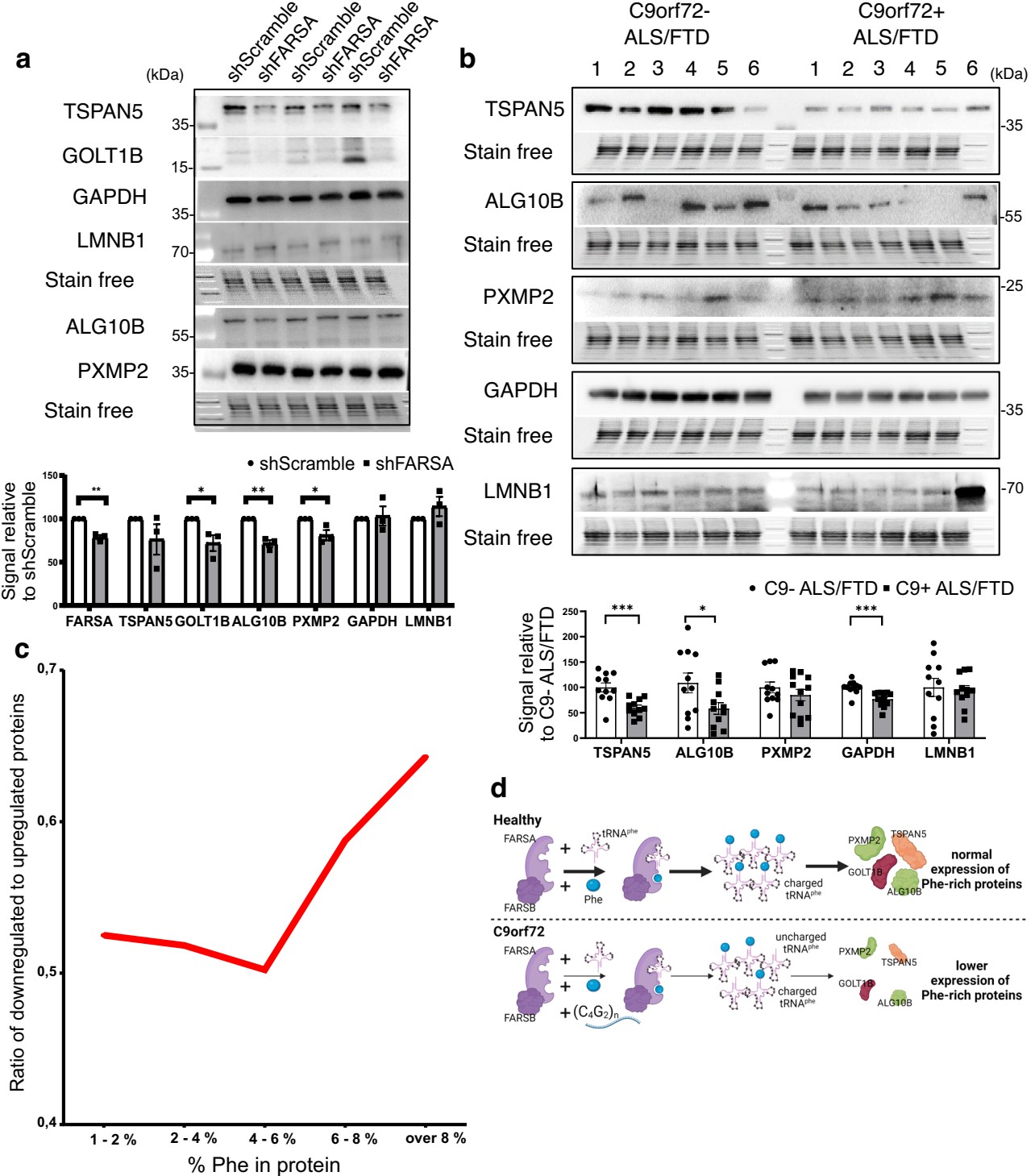

Rabbit mAb (5597 S, lot: D5A7, Cell Signaling Technology, Danvers, Massachusetts, USA, PLA 1:100); GOLT1B (PA5-103499 Invitrogen, Waltham, Massachusetts, ZDA; WB 1:1000, WB-tissue 1:500); GOLT1B (HPA055909: WB 1:1000), PABPC1 (P6246, lot: 015M4782V; ICC 1:250) (all from Sigma Aldrich, St. Louis, Misuri, ZDA). The following secondary antibodies were used: Alexa Fluor 488 (4408; dilution 1:1000), Alexa fluor 488 (A21202; dilution 1:1000), Alexa Fluor 555 (A31572; dilution 1:1000), Alexa Fluor 647 (A21447; dilution 1:1000) (all from Cell Signaling Technology, Danvers, Massachusetts, USA) StarBright Blue 520 (#12005866, #12005869), StarBright Blue 700 (#12004158, #12004161) Fluorescent Secondary Antibodies (all from Bio-Rad, Hercules, California, USA; dilution 1:5000); Peroxidase AffiniPure Goat Anti-Rabbit

IgG (H + L) (111-035-045), Anti-Mouse IgG (H + L) (115-035-044) (both from Jackson ImmunoResearch, West Grove, Pennsylvania, USA; dilution 1:5000). Control antibody for RNA immunoprecipitation assay was Normal Rabbit IgG (12-370, Millipore, Burlington, Massachusetts, USA).

### Biological resources

C9orf72 patient-derived fibroblasts were a kind gift from Dr. Don W. Cleveland (Ludwig Institute for Cancer Research, La Jolla, California, USA)[43]. Control fibroblasts were obtained from skin biopsies of healthy volunteers and approved by the Ethical Committee (EK45022009) of the Technische Universität Dresden, Germany and prepared as previously described[62]. They were cultured in high-

**Fig. 4 | Decreased expression of Phe-rich proteins in differentiated FARSA knockdown motor neuron-like NSC-34 cells and post-mortem cerebellum tissue of C9orf72 patients.** Western blot analysis revealed decreased expression of GOLT1B, PXMP2 and ALG10B Phe-rich proteins but not Phe-low proteins GAPDH and LAMB1 in (**a**) differentiated motor neuron-like NSC-34 cells. Expression levels relative to shScramble average (100%) are as follows: TSPAN5 (84.2 ± 9.8%, p-value = 0.198), PXMP2 (83.5 ± 5.0%, p-value = 0.044), GOLT1B (70.0 ± 8.5%, p-value = 0.017), ALG10B (74.4 ± 1.3%, p-value = 0.0023), GAPDH (109.6 ± 11.3%, p-value = 0.923) and LAMINB1 (125.3 ± 21.2%, p-value = 0.596). Statistics was done using nested t-test on 3 independent experiments. Expression levels of TSPAN5 and ALG10B were significantly reduced in (**b**) RIPA lysates extracted from frozen cerebellar gray matter of ALS/FTD cases with a C9orf72 mutation (C9 + ALS/FTD). Expression levels relative to controls (ALS/FTD cases without C9orf72 mutation (C9- ALS/FTD)) average (100%) were as follows: ALG10B (53.8 ± 10.8%, p-value (two-sided) = 0.048), TSPAN5 (60.3 ± 5.1%, p-value (two-sided) = 0.0009), GAPDH (76.7 ± 4.0%, p-value (two-sided) = 0.0004), LAMINB1 (94.1 ± 9.6%, p-value (two-sided) = 0.773). Statistics was done using unpaired Student's t-test on twelve C9 + ALS/FTD and 12 C9- ALS/FTD biologically independent samples Stain free images represent loading comparison for all proteins above the stain free image and are vertically compressed for design reasons. Molecular weight is marked on western blot pictures in kDa. Graphs present mean values ± s.e.m. with statistical significance labeled as *p < 0.05, **p < 0.01, ***p < 0.001. **c** Proteomic analysis indicates that the ratio of downregulated to upregulated genes, presented with the red line, increases with the increasing % of Phe in the proteins in C9orf72-patient derived lymphoblastoid cell lines. Mass spectrometry analysis was performed on six C9orf72 patient-derived and six control biologically independent samples for each sample once. **d** Schematic representation (created with Biorender.com) of results. Source data are provided as a Source data file.

**Table 1 | Summary of Phe-rich and Phe-low protein expression studies in evaluated cell models and human cerebellum post-mortem tissue**

|  | HEK293 FARSA KD | Differentiated FARS A KD NSC-34 | Lymphoblastoid cell lines | Human cerebellum |
|---|---|---|---|---|
| TSPAN5 | ↓* | n.s. | ↓* | ↓*** |
| GOLT1B | ↓* | ↓* | ↓* | n.d. |
| ALG10B | ↓* | ↓** | ↓* | ↓* |
| PXMP2 | ↓* | ↓* | ↓*** | n.s. |
| GAPDH | n.s. | n.s. | n.s. | ↓*** |
| LAMINB1 | n.s. | n.s. | n.s. | n.s. |

Reduction in protein expression (densiometric signal on WB) is indicated with downward arrow. Statistical significance is labeled as *p < 0.05, **p < 0.01, ***p < 0.001. n.s. represent non-significant results and n.d. not determined. Source data are provided as a Source data file.

glucose Dulbecco's modified Eagle's medium with GlutaMax (DMEM) supplemented with 20% fetal bovine serum (FBS) and 100 U/mL penicillin/streptomycin (P/S) (all from Gibco, Thermo Fisher Scientific, Waltham, Massachusetts, USA). C9orf72 patient-derived and control lymphoblastoid cell lines were prepared from patients segregating C9orf72 mutations as previously described[63]. They were cultured in RPMI 1640 (Gibco, Thermo Fisher Scientific, Waltham, Massachusetts, USA) supplemented with 15% FBS and 100 U/mL P/S. C9orf72 patient-derived (cell line CS52iALS-C9n6) and isogenic control (cell line CS52iALS-C9n6.ISOC3) induced pluripotent stem cells (iPSCs) were purchased from Cedars-Sinai Medical Center (LA, California, USA). These iPSCs were cultured on Matrigel (Corning Life Sciences, Glendale, Arizona, USA) in mTeSR1 medium (STEMCELL Technologies, Vancouver, Canada). HEK293T (ATTC, CRL-3216) and HEK293 (ATTC, CRL-1573) cells were cultured in DMEM supplemented with 10% FBS and 100 U/mL P/S. All cell lines were maintained in a humidified atmosphere containing 5% CO$_2$ at 37 °C. NSC-34 cells (Cedarlane Laboratories, CLU140) were cultured in DMEM supplemented with 10% FBS, 1% Sodium pyruvate and 100 U/mL P/S in non-differentiating conditions. All cell lines were confirmed to be mycoplasma-free by Eurofins Genomics (Luxembourg, EU).

The plasmid construct with 32×CCCCGG repeats containing the S1m aptamer on the 3′-end (pcDNA3.1-32×C4G2-S1m) and control constructs—a partial sequence (369 bp) of red fluorescent protein with the S1m aptamer on the 3′-end (pcDNA3.1-RFP-S1m) and a construct containing only the S1m aptamer (pcDNA3.1-S1m)—were previously described[64]. Plasmids with short 8x and 24x C4G2 repeats were cloned from previously described G4C2 repeats[15]. The plasmid construct PB-513B-1-GFP-MmPheT413G containing mutated FARSA (MmPheT413G) was obtained from dr. Tony Wyss-Coray lab and was previously described[38]. Sequence encoding for mutated FARSA was cloned into pcDNA 3.1. vector tagged with HA-tag. All the sequences were confirmed by sequencing (Microsynth AG, Switzerland, EU).

## Preparation of RNA constructs
For RNA preparation, all constructs were linearized at the restriction site on the 3′-end of the S1m aptamer. The constructs contained the T7 promoter for in vitro transcription to RNA, which was performed with the TranscriptAid T7 High Yield Transcription Kit (K0441, Thermo Fisher Scientific, Waltham, Massachusetts, USA). Single-strand Binding Protein from *Escherichia coli* (S3917, Sigma Aldrich, St. Louis, Missouri, USA) was added to the reaction (7.5 µg per 1 µg of DNA) to facilitate the transcription of GC-rich hexanucleotide repeats. The reactions were performed for 6 h at 37 °C. The integrity of the prepared RNA constructs was assessed with agarose gel electrophoresis.

## Tissue and cell extraction
Mouse brains obtained from *Mus Musculus* strain FVB/PyMT, age 2 y, female, were obtained under the approval of the Veterinary Administration of the Ministry of Agriculture and the Environment, Slovenia. Procedures for animal care and use were in accordance with the "PHS Policy on Human Care and Use of Laboratory Animals" and the "Guide for the Care and Use of Laboratory Animals" (NIH publication 86-23, 1996). Mouse brain tissue was processed and nuclear and cytoplasmic fractions were prepared as previously described[20]. Mouse brain tissue (300 mg per experiment) was homogenized in 5 mL of ice-cold buffer A [10 mM HEPES, 10 mM KCl, 1.5 mM MgCl2, protease inhibitors (Roche, Basel, Switzerland)] with glass pestle tissue grinder on ice. The homogenate was incubated on ice for 75 min and centrifuged for 10 min at 3000 × g at 4 °C. The pellet (nuclear fraction) was washed 3× with 1.5 mL of buffer A and centrifuged for 10 min at 3000 × g at 4 °C each time. The pellet was then re-suspended in 1.25 mL of RNA lysis buffer [50 mM HEPES, 400 mM KCl, 10 mM MgCl$_2$, 1% Igepal], incubated on ice for 10 min, and sonicated (30 s, 0.5 cycle, 90% amplitude). Both cytosol and nuclear fractions were centrifuged for 10 min at 16,000 × g at 4 °C, and supernatants were collected. The nuclear fraction was diluted 4× with dilution buffer [50 mM HEPES, 10 mM MgCl2, 0.33% Igepal CA-630]. The RiboLock RNase Inhibitor (Fermentas, Waltham, Massachusetts, USA) was added to both fractions at a 1:400 (v/v) ratio. Aliquots for western blot lysate controls were taken at this point.

For interaction validation studies with RNA pull-down, frontal cortex (300 mg per experiment) from a control case (male, 63 y) was prepared without fractionation. Fresh-frozen post-mortem tissue was obtained from the Brain Bank associated with the DZNE and University Hospital of Tübingen (IEC Project Numbers 252/2013BO1 and 386/2017BO1). The tissue was homogenized in RNA lysis buffer as previously described and then diluted 4× with dilution buffer. Same was done for HEK293 cells. HEK293T cells were previously transfected with pcDNA3.1.-mFARSA-HA construct using PolyJet reagent (SignaGen,

Frederick, USA), lysed with RNA lysis buffer after 24 h and 4x diluted in dilution buffer.

For Phe-rich protein expression analysis, the study cohort consisted of 12 cases with a *C9orf72* repeat expansion mutation presenting with ALS ($n = 4$), FTD/ALS ($n = 5$) or FTD ($n = 3$) and 12 control cases with TDP-43 pathology in the absence of a *C9orf72* repeat expansion mutation presenting with ALS ($n = 7$), ALS/FTD ($n = 3$) and FTD ($n = 2$). Details for cases are provided in Supplementary Table 4. Cerebellum samples were homogenized in RIPA buffer (50 mM Tris-HCl, 150 mM NaCl, 1% (v/v) octylphenoxy poly(ethyleneoxy)ethanol (IPEGAL), 5 mM EDTA, 0.5% (w/v) sodium deoxycholate, 0.1% (w/v) sodium dodecyl sulfate (SDS), pH 8.0) at 1 g / 2 ml ratio. Lysates were passed through 18- and 21- Gauge needles and sonicated to shear nucleic acids. Cellular debris was removed by centrifugation for 2 min at 3000 × *g* at 4 °C and supernatant collected as RIPA lysate.

### Knockdown assay
NSC-34 cells were differentiated in differentiating medium (DMEM F12 + 0,5% FBS + 1 x MEM + 1% P/S + 10 μM retinoic acid) modified according to[65] seven days prior to lentiviral transduction with second generation lentiviruses prepared in HEK293T cells. At 70% confluency HEK293T cells were co-transfected using PolyJet (SignaGen Laboratories, Frederick, USA) reagent with pMD2.G (Addgene #12259), psPAX2 (Addgene #12260) and suitable shFARSA RNA (Sigma-Aldrich mission plasmids 07082112MN: 1584s21c1 for NSC34 and 1357s1c1 for HEK293) or shScramble (Addgene #1864) in 1:2:3 ratios according to the manufacturer's instructions. 6 h later medium was exchanged with HEK293 or NSC-34 growth medium. After 48 h the supernatant was collected and filtered through a 0.45 um cellulose acetate membrane and added to HEK293 (1:3 dilution in growth medium) or differentiated NSC-34 (1:1 dilution in differentiating medium) cells. After 24 h the medium was changed and cells were incubated for additional 48 h when they were fixed for IF or collected for WB and qPCR analysis.

### RNA pull-down assay
For the RNA pull-down assay, 100 pmol of RNA construct per reaction diluted in RNA-binding buffer [50 mM HEPES (pH 7.4), 100 mM KCl, 10 mM MgCl$_2$, 0.5% Igepal CA-630] was incubated with 50 μL of streptavidin beads (Promega, Madison, Wisconsin, USA) for 30 min at room temperature (RT). RiboLock RNase inhibitors were used throughout the experiment (1:100). The supernatant was removed, and the beads were washed 2× with 500 μL of RNA-binding buffer; tRNA from baker's yeast (R8508, Sigma Aldrich, St. Louis, Missouri, USA) was added to cytoplasmic and nuclear mouse tissue fractions and to lysates from human postmortem brain tissue. The beads with bound RNA were rotated with 1 mL of cytoplasmic or nuclear fraction for 4 h at 4 °C. The beads were washed 4× with 500 μL of RNA-binding buffer. Elution was performed with 3U RNase I (EN0601, Thermo Fisher Scientific, Waltham, Massachusetts, USA) for 10 min at 37 °C. Cytoplasmic and nuclear mouse tissue fractions (two independent experimental repeats), lysates from human postmortem brain tissue (two independent experimental repeats), HEK293 (three independent experimental repeats) and HEK293T (two independent experimental repeats) cell lysate were used for RNA pull-down assay.

### Mass spectrometry
Mouse brain samples from RNA-pull down assay were prepared in 2× sodium dodecyl sulfate (SDS) loading buffer and 0.2 M dithiothreitol (DTT) and incubated for 5 min at 95 °C. The samples were then separated with SDS-PAGE, and the gels were silver stained. The selected bands were cut out and digested using a ProGest digestion robot (DigiLab)[66]. Briefly, the gel bands were diced, put into the pink reaction plate, and destained with destaining solution (15 mM potassium ferricyanide, 50 mM sodium thiosulfate) for 15 min. After washing with 50 mM ammonium bicarbonate (ABC) then acetonitrile (ACN) two

times, the proteins were reduced with 10 mM DTT at 37 °C for 10 min and alkylated with 50 mM iodoacetamide for 15 min at RT. This was followed by two washes with 50 mM ABC and ACN. The proteins were digested with trypsin (Pierce, Thermo Fisher Scientific, Waltham, Massachusetts, USA) overnight at 37 °C. The digested peptides were extracted from the gel pieces by sequentially adding 25 mM ammonium bicarbonate, acetonitrile, and 10% formic acid and then collected into a blue collection plate. The peptides were freeze-dried and resuspended in 2% ACN in 0.05% trifluoroacetic acid (TFA). The samples were separated with nano-flow liquid chromatography on a reverse-phase column (C18 PepMap100, 3 μm, 100 Å, 50 cm; Thermo Fisher Scientific) using 80 min gradient (0–37 min, 4% B – 30% B; 37–40 min, 30% B – 40% B; 40–45 min, 40% B – 99% B; 45–50 min, 99% B; 50–80 min, 4% B. Solvent A = 2% ACN, 0.1% FA in H2O; B = 80% ACN, 0.1% FA in H$_2$O). Spectra were collected from an Orbitrap mass analyzer (Q Exactive HF, Thermo Fisher Scientific, Waltham, Massachusetts, USA) using a full ion scan mode over the *m/z* range 350–1600 and resolution 60,000. For each full scan, MS/MS scans were performed on the 15 most abundant ions using higher-energy collisional dissociation (HCD) and measured with Orbitrap (resolution 15,000) with dynamic exclusion enabled (40 s).

Data were analyzed with Proteome Discoverer (version 2.1, Thermo Fisher Scientific) against UniProt/SwissProt mouse database (version 2017_01, 16,844 protein entries) using Mascot (version 2.6.0, Matrix Sciences). The search results were loaded into Scaffold (version 4.8.7, Proteome Software). Carboxyamidomethylation of cysteine was used as a fixed modification, and oxidation of methionine was used as a variable modification. The mass tolerance was set at 10 ppm for the precursor ions and at 20 mmu for fragment ions. Two missed cleavages were allowed. The search results were loaded into Scaffold software (version 4.8.7, Proteome Software), and protein and peptide probabilities were calculated. Assignments were accepted with >95.0% peptide probability, >99.0% protein probability, and a minimum of two peptides per protein. The mass spectrometry was performed on one sample from each condition as a screening step for possible binding proteins. Protein candidates with correct molecular weight as indicated on the SDS-PAGE gels, spectral count >20 and fold change >3 between C4G2 and S1m control were further validated by Western blot.

Twelve independent biological samples: six C9orf72-patient derived and six control lymphoblastoid cell lines were used for MS analysis of whole proteome expression in one technical replicate. $1 \times 10^6$ cells were washed 3 times with ice cold PBS and collected in RIPA buffer (50 mM Tris/HCl, pH 8; 150 mM NaCl; 0,5 Mm Sodium deoxycholate; 0,1% SDS and 1% Triton X-100). They were centrifuged at 12,000 × *g* for 10 min and supernatant was collected for MS analysis.

Protein concentration was measured using BCA method and 50ug of proteins from each sample were used. Protein samples were denatured in 6 M urea / 2 M thiourea and reduced by 10 mM DTT at 37 °C for 1 h. Then samples were alkylated using 50 mM iodoacetamide for 1 h in the dark at room temperature. After acetone precipitation over night at −20 °C, the protein pellets were resuspended in 100 mM TEAB (pH = 8.5) and digested using trypsin / lysC (Promega, protein: enzyme ratio = 25:1) at 37 °C overnight. The digestion was stopped by 1% trifluoroacetic acid and peptides were purified using C18 cartridges on Bravo AssayMAP robot (Agilent). The cleaned peptides were dried and resuspended in 2% acetonitrile (ACN), 0.1% formic acid (FA) and separated by nano-flow HPLC (U3000 RSLCnano, Acclaim PepMap100 C18 column, 75 μm × 50 cm, Thermo Fisher Scientific) using 2 h gradient and analysed using Orbitrap Fusion Lumos MS with FAIMS Duo Pro (3 different CV: −40 V, −50 V, −60 V).

RAW data were searched against human protein database (UniProt/SwissProt Release 2022_01, 20376 protein entries) combined with FBS contamination database (249 protein entries) using Mascot algorithm (version 2.6.0, Matrix Science) and Proteome Discoverer

(version 2.4.1.15, Thermo Fisher Scientific) with the following parameters: trypsin was used as enzyme and 2 missed-cleavages was allowed; precursor mass tolerance was set at 10 ppm and fragment mass tolerance was 0.8 Da; Carbamidomethylation on cysteine was set as static modification and oxidation on methionine as dynamic modification. Protein table was exported for further analysis (Supplementary Data 1).

Analysis of ratio between downregulated and upregulated proteins in comparison to Phe-content was performed on all proteins that had log 2(Fold Change) of at least 0,15 and that we know the percentage of Phe (1512 proteins) (Supplementary Data 1). Each protein was associated with the % of Phe in its sequence according to[39]. We calculated number of downregulated and upregulated proteins in each interval (0–2%, 2–4%, 4–6%, 6–8%, over 8%).

The proteomics data were deposited to the ProteomeXchange Consortium via the PRIDE[67] partner repository with dataset identifier PXD024527 and PXD042474.

## RNA immunoprecipitation

For RNA immunoprecipitation (RIP) assay C9orf72 patient-derived fibroblasts were used. Cells were washed twice with cold PBS and lysed in ice-cold lysis buffer [50 mM Tris-HCl, pH 7.4, 100 mM NaCl, 1% Igepal CA-630, 0.1% SDS, 0.5% sodium deoxycholate]. Protease inhibitors (PI) and RiboLock RNase Inhibitor were added to the lysis buffer. Cells were incubated in lysis buffer at 4 °C for 30 min, sonicated (3 × 10 s, 0.5 cycle, 90% amplitude), and centrifuged. Supernatant was transferred to a new tube and 3 µg of either FARSA antibodies or control Normal Rabbit IgG antibodies were added to the lysate and incubated over night at 4 °C. Dynabeads Protein A (10001D, Thermo Fisher Scientific, Waltham, Massachusetts, USA) were washed twice with lysis buffer and then incubated with the lysate for 30 min at 4 °C. Three washes were performed with wash buffer [50 mM Tris-HCl, pH 7.4, 1 M NaCl, 1% Igepal CA-630, 0.1% SDS, 0.5% sodium deoxycholate]. 2×SDS loading buffer with 200 mM DTT was added to 10% of the beads and incubated at 95 °C for 5 min. RNA was isolated from the RIP reaction with TRIzol. For cDNA synthesis SuperScript IV Reverse Transcriptase (18090010 Thermo Fisher Scientific, Waltham, Massachusetts, USA) was used. Reaction conditions and primers for antisense RNA were used as previously described[17]. Detection was carried out with qPCR analysis using FastStart Universal SYBR Green Master (Rox) (Roche, Switzerland, EU) and a StepOnePlus Real-Time PCR system (Applied Biosystems, USA). Cycling conditions were: 95 °C for 10 min, 45 cycles at 95 °C for 10 s, and 62 °C for 35 s, followed by a melting curve analysis. Primer PCR efficiencies were at least 80% and a single melting peak was present for each primer pair. Three independent experiments on C9orf72 patient-derived fibroblasts were performed.

## Western blot analysis

Protein samples were prepared in RIPA buffer (50 mM Tris/HCl, pH 8; 150 mM NaCl; 0,5 mM Sodium deoxycholate; 0,1% SDS and 1% Triton X-100 + protease inhibitors) and 2×SDS loading buffer with 200 mM DTT, incubated at 95 °C for 5 min and separated on 12% SDS precast gels (Invitrogen, Carlsbad, California, USA) or 12% stain free gels (Bio-Rad, Hercules, California, USA) at 125 V. The samples were wet-transferred onto nitrocellulose membrane (GE Healthcare, Chicago, Illinois, USA) at 200 mA for 90 min or semi-dry transferred (Bio-rad, Hercules, California, USA). Blocking was performed for 1 h at RT in blocking solution (5% skim milk in TBS with 0.05% Tween-20 (TBST) or 2.5% skim milk + 1% BSA in TBST for TSPAN5, GOLT1B, ALG10B and GAPDH), incubations with the primary and secondary antibodies were performed as suggested by the manufacturer. Clarity Max Western ECL Substrate (Bio-Rad, Hercules, California, USA) was used for signal detection. The GelDoc System ImageLab software (both from Bio-Rad, Hercules, California, USA) and imageJ (https://imagej.nih.gov/) were used for image acquisition and densitometric analysis, respectively.

## RNA fluorescent in situ hybridization (RNA-FISH) and immuno-fluorescence staining (IF)

Cells were fixed on coverslips with 4% paraformaldehyde for 15 min, washed with PBS, and permeabilized with 0.1% Triton in PBS for 5 min. RNA-FISH was performed as previously described[20,64]. Shortly, the coverslips were incubated in pre-hybridization solution [40% formamide, 2× saline sodium citrate (SSC)] for 20 min at RT and then in hybridization solution [2×SSC, 100 µg/mL tRNA (R8508, Sigma), 10% dextran sulfate, 25% formamide] for 20 min to 1 h at 59 °C. The locked nucleic acid (LNA) probes were dissolved and heated in hybridization solution at 95 °C for 5 min. The coverslips were then incubated with 40 nM LNA probes at 59 °C overnight. Stringency washes were as follows: 1×5 min in 2×SSC, 0.1% Tween-20 at RT and 3×10 min in 0.1×SSC at 59 °C. This was followed by the IF protocol. Coverslips were blocked in 10% goat serum (Sigma Aldrich, St. Louis, Missouri, USA) or in 3% biotin-free bovine serum albumin (Carl Roth, Germany, EU) for 30 min at RT. Incubation with primary antibodies diluted in blocking solution was performed for 1.5 h at RT and incubation with secondary antibodies was performed for 1 h at RT. The coverslips were mounted using ProLong Gold Antifade reagent (P36930, Thermo Fisher Scientific, Waltham, Massachusetts, USA). The following LNA probes were purchased from Exiqon (Qiagen, Germany, EU): /5TYE563/GGGGCCGGGGCCGGGG, /5Biosg/GGGGCCGGGGCCGGGG. For RNA-protein fluorescence co-localization, the ImageJ Plot profile function was used.

## RNA-protein proximity ligation assay (PLA)

The RNA-protein PLA was optimized from previously described protocols[43,68]. The hybridization solution for RNA-ISH contained 50% formamide, 5×SSC, 1×Denhardt's solution, 0.1% Tween, 0.1% Chaps, 5 mM EDTA, 0.1 mg/mL tRNA, and 0.1 mg/mL heparin. The steps for RNA-ISH were the same as for RNA-FISH. Coverslips were then blocked in 3% biotin-free bovine serum albumin (Carl Roth, Germany, EU) for 30 min at RT. Incubation with primary antibodies against biotin (detecting LNA probe) and against target antibody was performed for 1.5 h at RT. PLA antibody probes (Duolink, Sigma Aldrich, St. Louis, Missouri, USA) were incubated on slides for 1 h at 37 °C. The PLA protocol from here onward was performed according to the manufacturer's instructions. The same antibodies and dilutions were used as for IF. For the PLA signal count, the ImageJ find maxima function was used on each slice of the z-stack, and counted signals were summed for each cell through the stacks.

## Confocal microscopy imaging

Images were acquired with a Zeiss LSM 710 inverted confocal laser scanning microscope with a Plan-Apochromat 63× and 1.4 NA M27 oil immersion objective using immersion oil (Carl Zeiss). DAPI, Alexa Fluor 488, Alexa Fluor 555, Alexa Fluor 647 and 5TYE563 were excited at 405, 488, 543, 647 and 543 nm, respectively. The zoom factor was set to 1–4×, and X- and Y-scanning sizes were each 1024 pixels. All other images except for HEK293 IF were acquired as z-stacks, and the z-scanning size was 0.979–2 µm.

## In vitro aminoacylation assay

Recombinant FARSA and FARSB proteins from Drosophila were expressed in the E. coli strain Rosetta (Novagen) and then purified[69]. For this, the Drosophila α-PheRS (FARSA) cDNA was cloned with a His tag at the N-terminal end into the pET-28a plasmid expression vector (Novagen). The α-PheRS (FARSB) cDNAs was cloned into the pET LIC (2A-T) plasmid (Addgene). The two subunits were then co-expressed in the E. coli strain Rosetta with isopropylthiogalactoside (IPTG, 1 mM) induction at 25 °C for 6 h. The IPTG was added when the OD600 of bacteria reached 0.6-0.8. Proteins were purified with Ni-NTA affinity resin (Qiagen). The aminoacylation assay protocol from Lu and colleagues was then followed[36]. This assay was performed at 25 °C in a 100 µl reaction mixture containing 50 mM Tris-HCl pH 7.5, 10 mM

MgCl$_2$, 4 mM ATP, 5 mM β-mercaptoethanol, 100 µg/ml BSA, 4.375 µg PheRS/FARS protein, 3 U/ml *E. coli* carrier tRNA, 5 µM [$^3$H] L-Phe and 1 µM tRNA$^{Phe}$ from brewer's yeast (Sigma, US). The inhibitory effect of RNA C4G2 or RFP was tested by pre-incubating for 10 min each RNA at two different concentrations (0.4 ng/uL and 4 ng/uL) in the master mix before adding tRNA$^{Phe}$ and [$^3$H] L-Phe to start the reaction. In each experiment, a 10-µl aliquot was removed at six different incubation time points, spotted on a Phe saturated Whatman filter paper disc (the discs were soaked in 0.2 M Phe solution for at least 2 h at RT) and washed three times with ice-cold 5% trichloroacetic acid and once with ice-cold ethanol. A blank paper disc without spotting and another with spotting the enzyme-free reaction were used for detecting background signals. After filter discs were dried, they were immersed in PPO Toluol (Sigma, US) solution in plastic bottles and the radioactivity was measured by scintillation counting.

## tRNA aminoacylation assay in cells

The assay for determining tRNA aminoacylation levels has been described previously in Loayza-Puch[37]. Lymphoblastoid cells (5–10 million) were harvested, washed with cold PBS, and resuspended in ice-cold 0.3 M sodium acetate/acetic acid (NaOAc/HOAc; pH 4.5). The RNA was isolated using TRIzol and resuspended in 10 mM NaOAc/HOAc (pH 4.5). The RNA samples (5 µg) were split into two halves that were either oxidized with 50 mM sodium periodate or incubated with 50 mM sodium chloride in 100 mM NaOAc/HOAc (pH 4.5) for 15 min at RT. The reactions were quenched with 100 mM glucose for 5 min at RT and purified in G25 columns (GE Healthcare, Chicago, Illinois, USA) following manufacturer's instructions. Purity of the RNA was measured by using Nanodrop 2000, with 260/280 ratio ~2. The RNA was spiked with 7.3 ng yeast tRNA$^{Phe}$ (R4018, Sigma Aldrich, St. Louis, Missouri, USA) per 1 µg of RNA. Deacylation of tRNA was performed in 50 mM Tris-HCl (pH 9) at 37 °C for 30 min. The RNA was precipitated, resuspended in diethyl pyrocarbonate (DEPC)-treated water, and ligated to the 3'adaptor (5'-/5rApp/TGGAATTCTCGGGTGCCAAGG/3ddC/-3') with T4 RNA ligase 2 truncated (M0242, New England Biolabs, Massachusetts, USA) for 2 h. Then, the RNA was purified with phenol-chloroform extraction and reverse transcribed with TGIRT™-III enzyme (InGex, Missouri, USA) and the primer 5' GCCTTGGCACCCGAGAATTCCA 3'. The RT was performed in 20 µL reactions. First, 5×TGIRT buffer, template RNA (300 ng), RT primer (1 µM), MgCl2 (50 mM), of RNase free water were mixed, heated to 75 °C for 3 min, and incubated on ice for 1 min. Then TGIRT-III enzyme, RNasIn PLUS (Promega Madison, Wisconsin, USA), and 100 mM DTT were added to the reaction mix and incubated at RT for 20 min. After addition of dNTP mix (500 µM) the reaction mix was incubated at 57 °C for 60 min. The absence of DNA was assessed with a no-reverse transcription assay. The synthesized cDNA was diluted 1:5 and qPCR cycling conditions were: 95 °C for 10 min, 45 cycles at 95 °C for 10 s, and 62 °C for 35 s, followed by a melting curve analysis. Primer efficiencies were at least 80% and a single melting peak was present for each primer pair. The following primers were used: forward Phe(GAA)-tRNA primer GCCGAAATAGCTCAGTTGGGAG; forward yeast (spike-in) Phe-tRNA primer GCGGAYTTAGCTCAGTTGGGAGAG; Asn(GTT)-tRNA GTCTCTGTGGCGCAATCGGT, Leu(CAA)-tRNAGTCAGGATGGCCGAGTGGTCTA, Leu(CAG)-tRNA GTCAGGATGGCCGAGCGGTCTA, Pro(CGG)-tRNA GGCTCGTTGGTCTAGGGGTATG, and reverse primer GCCTTGGCACCCGAGAATTCCA.

## Amino acid mass spectrometry analysis

Cell pellets were resuspended in 250 µL of distilled water and sonicated for 2 min using a Labsonic M probe sonicator (Stratorius Stedim Biotech, Göttingen, Germany). After centrifugation at 16,800 × *g* for 2 min at RT, supernatants were used for amino acid analysis with the aTRAQ Kit for Physiological Fluids (Sciex, Massachusetts, USA). Samples were prepared as described in the protocol. Briefly, after precipitation of proteins with sulfosalicylic acid, the amino acids were derivatized with the aTRAQ reagent. Derivatization was stopped with hydroxylamine, and the aTRAQ internal standard was added. The samples were analyzed in multiple reaction monitoring mode with the AB Sciex 3200 Qtrap tandem mass spectrometer (Sciex, Massachusetts, USA), Perkin Elmer Series 200 HPLC system (Perkin Elmer, Massachusetts, USA). Chromatographic separation of derivatized samples was achieved at 50 °C on a C18 4.6 × 150 mm column from the aTRAQ kit. The mobile phase A consisted of water (LiChrosolv®, Merck, Darmstadt, Germany), and the mobile phase B consisted of methanol (LC-MS Reagent, J.T.Baker, Deventer, The Netherlands); both contained 0.1% formic acid and 0.01% heptafluorobutyric acid (both from Sciex, Massachusetts, USA).

## Phe-rich and stress related protein mRNA expression level analysis

Whole RNA was isolated using TRIzol from six control and six C9orf72 patient-derived lymphoblastoid cell lines, three independent knockdown experiments with three technical replicates of shScramble and shFARSA treated HEK293 and differentiated NSC-34 cell lines 3 days post transduction. 3 µg of total RNA was treated with DNaseI (Roche, Basel, Switzerland) and incubated for 30 min at 37 °C and 10 min at 75 °C. RNA was cleaned using phenol-chloroform extraction. cDNA was synthesized with High capacity cDNA RT kit (Applied biosystems, Waltham, Massachusetts, USA). Reverse transcription was performed at 25 °C for 10 min followed by 120 min on 37 °C and 5 min at 85 °C. The absence of DNA was confirmed with a no-reverse transcription assay. cDNA was diluted 1:50 and qPCR analysis were performed with cycling conditions: 95 °C for 10 min, 45 cycles at 95 °C for 15 s, and 58 °C or 60 °C for 60 s, followed by a melting curve analysis. Primer efficiencies were at least 90% and a single melting peak was present for each primer pair. Primers *ALG10B*-2; *GAPDH*-1; *GOLT1B*-1; *PXMP2*-2; *TSPAN5*-2; *ACTB*-1 were obtained from Sigma Aldrich (St. Louis, Missouri, USA), KiCqStart SYBR Green Primers. Primers for evaluating stress response listed in Supplementary Table 5 were purchased from Kemomed (Ljubljana, Slovenia).

## Click chemistry analysis of phenylalanine abundance in the proteome

Stable HEK293T cell line expressing mutated FARSA protein that enables incorporation of azido-Phe into proteome of the cell was established using construct for piggybac transposase and PB-513B-1-GFP-MmPheT413G. Both plasmids were co-transfected into HEK 293T cells using Lipofectamine 3000 and incubated for 72 h before puromycin selection (10 µg/µl). The presence of the construct was observed as continuous expression of GFP.

Cells were transfected with pcDNA3.1 construct encoding S1m, RFP-S1m and 32×C4G2-S1m using Polyjet reagent (SignaGen Laboratories, Frederick, USA). 24 h later azido-Phe (JennaBioscience, Jena, Germany) was added to a final concentration of 125 µM. After 24 h incubation cell lysates were collected in lysis buffer (1% SDS in PBS with protease inhibitors), sonicated 3 × 10 s at amplitude of 80%. 25 µg of protein was added to the copper-catalyzed Click reaction kit (Jena, Germany). The reaction was done according to the manufacturer's protocol. Alexa 555 alkyl (Invitrogen, Waltham, Massachusetts, USA) was used for staining and imaged on GelDoc System (Bio-Rad, Hercules, California, USA). Gels were stained with colloidal Coomassie dye G-250 (Gelcode Blue stain reagent, Thermo Fisher Scientific, Waltham, Massachusetts, USA) according to the manufacturer's protocol and imaged on GelDoc System (Bio-Rad, Hercules, California, USA).

## Quantification and statistical analysis

For the antisense RNA foci-FARSA protein co-localization studies, three C9orf72 patient-derived fibroblast cell lines were used, and the

RNA-FISH/IF experiments were performed independently at least two times for each cell line. The numbers of cells and foci analyzed in each cell line were at least 40 and 250, respectively.

For RNA-protein PLA, three C9orf72 patient-derived and three control fibroblast cell lines, three C9orf72 patient-derived and three control lymphoblastoid cell lines, and one C9orf72 patient-derived and one control iPS cell line were used. The experiments were performed independently at least two times for each cell line. The numbers of cells analyzed in each independent experiment for fibroblasts were at least 100, for lymphoblastoid cells at least 200, and for iPSCs at least 300.

For the tRNA aminoacylation assay, four different C9orf72 patient-derived and four different control lymphoblastoid cell lines were used. The experiment was performed in four independent replicates and three technical replicates were performed for the qPCR analysis of each experiment. The ddCt method was used for qPCR data analysis and the yeast tRNA$^{Phe}$ spike-in was used for normalization.

The amino acid content was determined in six different C9orf72 patient-derived and six different control lymphoblastoid cell lines. The experiment was performed once for each cell line. The results were compared between C9orf72 patient-derived and control lymphoblastoid cell lines.

The expression of FARSA was determined in six different C9orf72 patient-derived and six different control lymphoblastoid cell line lysates. The experiment was performed in two technical replicates for each cell line. The results were compared between C9orf72 patient-derived and control lymphoblastoid cell lines.

Phe incorporation experiments using click chemistry were repeated four times. Expression level was measured using ImageJ and normalized to whole protein expression determined by colloidal Coomassie dye G-250. Data were reported relative to S1m.

Densitometry signals for each Phe-rich protein were always normalized to whole protein loading from stain free membrane. In knockdown experiments four (HEK293) or three (NSC-34) independent experiments with three technical replicates were done. Each independent experiment was analyzed on one membrane (controls + KD samples). Each value was reported relative to shScramble average and ratios were compared between blots. Statistical significance was calculated using nested t-test analysis in GrafpadPrism version 9.5.0. for Windows, GraphPad Software, San Diego, California USA, www.graphpad.com on all included technical replicates and biological repeats. Six control and six C9orf72 patient-derived lymphoblastoid cell lines (each representing one biological repeat) were compared on the same membrane. Experiments were repeated in two technical repeats. Unpaired Student's t-test was performed between 6 biological repeats (significance marked on graphs) and additional statistical analysis was done between the dependent technical repeats with mixed model analysis in R to eliminate technical variations. Two models were constructed for each dependent variable and their parameters estimated based on the western blot densitometry data. In the first model (model A) experiment (first/second experiment) was a fixed factor and each lymphoblastoid cell line was a random factor. In the second model (model B) additional parameter to determine C9orf72 patient-derived or control group was included and used as a fixed factor. Model B was compared against the model A using likelihood ratio test to assess differences between the C9orf72 patient-derived and control cells with respect to the mean values of the protein expression levels.

Twenty-four human post mortem tissue samples were analyzed on two membranes: 6 C9orf72-mutation positive samples and 6 C9orf72-mutation negative samples on each. Each sample was normalized to C9orf72-mutation negative samples average. Ratios were compared between membranes.

qPCR analysis of Phe-rich mRNA expression level was done in three technical repeats for each sample using six control and six C9orf72 patient-derived lymphoblastoid cell lines. qPCR analysis of stress related proteins was done on three knockdown experiments with three replicates for each sample. Changes in expression levels were calculated using relative ΔΔcq qPCR analysis and normalized to ACTB transcript.

Statistical significance for previously described experiments unless otherwise specified was determined with the unpaired Student's t-test in Microsoft Excel 2010. A p-value (two-sided) of <0.05 was considered significant. All values are presented as means ± standard errors of the mean (s.e.m.), and statistical significance is indicated with * ($p < 0.05$), ** ($p < 0.01$), or *** ($p < 0.001$).

### Reporting summary
Further information on research design is available in the Nature Portfolio Reporting Summary linked to this article.

## Data availability
The mass spectrometry proteomics data from RNA pull-down experiment (*Mus musculus)* generated in this study have been deposited in the ProteomeXchange Consortium via the PRIDE partner repository database and are available through the accession code PXD024527. The mass spectrometry proteomics data from human lymphoblastoid cell lines generated in this study have been deposited in the ProteomeXchange Consortium via the PRIDE partner repository database and are available through the accession code PXD042474. Source data are provided with this paper.

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

## Acknowledgements

We thank Dr. Don W. Cleveland (Ludwig Institute for Cancer Research, La Jolla, CA) for providing C9orf72 patient-derived fibroblasts, Dr. Hannes Glaß for the help with patient-derived cells and Klementina Polanec and Vanja Čuk for technical help. This work was supported by the Slovenian Research Agency (Javna Agencija za Raziskovalno Dejavnost RS) [grant numbers: J3-3065, J3-4503, J7-3153, P4-0127, N3-0141, J7-9399, and J3-9263] and the International Centre for Genetic Engineering and Biotechnology (ICGEB) [grant number: CRP/SVN19-03]. Funding for open access charge: [Slovenian Research Agency]. This work was supported, in part, by the the NOMIS foundation to A.H. A.H. is supported by the Hermann und Lilly Schilling-Stiftung für medizinische Forschung im Stifterverband. M.T.H., B.S. and their work were supported by the Novartis Foundation for medical-biological Research [#18A050], the Swiss National Fund [grant number: 31003A_173188], and the University/Canton of Berne.

## Author contributions

Conceptualization: M.M.Č., U.Č., B.R.; Methodology: M.M.Č., U.Č., X.Y., M.M., B.R.L., M.T.H., B.R.; Validation: M.M.Č., U.Č.; Formal analysis: M.M.Č., U.Č., X.Y., B.S., B.R.L.; Investigation: M.M.Č., X.Y., B.R.; Resources: M.M., M.N., B.R.L., A.H., B.S., G.R., B.R.; Revision: U.Č., M.M.Č.; Writing - original draft: M.M.Č.; Writing - review & editing: M.M.Č., U.Č., X.Y., M.N., B.R.L., G.R., A.H., B.S., B.R.; Supervision: B.R.; Project administration: B.R.; Funding acquisition: A.H., B.S., B.R.

## Competing interests

The authors declare no competing interests.
