## [Peer Review File · Nature Communications]

Phenylalanine-tRNA aminoacylation is compromised by ALS/FTD-associated C9orf72 C4G2 repeat RNAREVIEWER COMMENTS

Reviewer #1 (Remarks to the Author):

In this interesting manuscript the protein interactors with long (30x) C4G2 repeats are identified. The interest of this exercise is identifying the targets that may be linked to the pathologies caused by such repeats.

Convincingly, PheRS is identified as one of such interactors, and the binding of C4G2 repeats to this enzyme is shown to modestly reduce tRNA aminoacylation. As a logical consequence of this inhibition, the authors show that the translation of Phe-rich transcripts is particularly affected because the amounts of freshly-translated Phe-rich proteins are clearly diminished with the C4G2 repeats present. This suggests that an important factor in C9orf72-related pathologies might be compromised protein synthesis machinery.

Although the data presented is clear, my concern with the manuscript is that it does not analyze in depth what is the functional consequence of this inhibitory effect over PheRS. In the discussion, the authors refer to other examples where compromised ARS function has an impact on fidelity, but the possibility of mistranslation at Phe codons caused by C4G2 repeats is not explored. Similarly, the fascinating sensitivity of neuronal tissues to translation defects is mentioned, but whether neuronal cells are more sensitive than other cells to C4G2 repeats is not analyzed.

Finally, the relative importance of PheRS inhibition with respect to the rest of C4G2 interactions is not measured, although it would be relatively simple to silence PheRS expression and look for the emergence of disease-related phenotypes.

Reviewer #2 (Remarks to the Author):

The manuscript of Malnar et al describes new aspects of the toxicity caused by the C4G2-repeat expansion in the C9orf72 gene causing AFD/ALS phenotype. The authors focus on studying the cytoplasmic interactions of the C4G2 antisense repeat RNA, a disease aspect that is underresearched, and discover that one of the binding partners is the phenylalanyl-tRNA synthetase (FARS). The neomorphic interaction is detected in C9orf72-derived fibroblasts, lymphoblasts, and iPSCs and is thus a ubiquitous finding. This interaction is differential, as it is not present when probing against other ARS, such as aspartyl- leucyl- and glutamyl-prolyl-tRNA synthetases. Further detailed biochemical studies in vitro and in different cellular model systems reveal a direct interaction between the alpha-subunit of FARS and the extended C4G2-transcript that has functional consequences on the FARS holoenzyme. The aminoacylation activity of FARS is significantly reduced both in in vitro aminoacylation assays and in total lysates from lymphoblasts of C9orf72-patients. Furthermore and importantly, the impaired aminoacylation activity of FARS has functional consequences on the global proteome of both HEK293 cell models expressing extended C4G2 repeat -mRNA and C9orf72 lymphoblasts. The authors demonstrate both in HEK293 cell models and C9orf72-lymphoblasts underrepresentation of Phe-rich proteins in the proteome (~20% reduction in their expression) at the background of normal phenylalanine amino acid cellular levels. Their hypothesis is that these proteome changes might have negative implications for the development of the C9orf72-related pathologies.

From a biochemical point of view, this is a very solid study that makes an original contribution to the field of both protein synthesis and neurodegeneration. The findings are original and somehow unexpected, but the provided evidence makes them well justified.

From a mechanistic perspective, the impact of the findings – neither for general cell biology nor for

neuronal biology - is unclear as no experimental data are provided nor hypotheses are raised. The findings pose several questions that await their answer before we can judge their contribution to enlightening the C9orf72-related neurodegeneration. It would be important that the authors provide experimental data or at least discuss their view on how their findings would be causally related to C9orf72 brain pathology in the patients. Some of the appealing open questions are: Why do neurons only suffer from the decrease of Phe-rich proteins while this phenomenon was detected in non-neuronal cell populations of the patients too? Is there a causal relationship between the partial loss of FARS activity and the C9orf72-neurodegeneration? The authors provide experiments in iPSC cells from patients, but it would be imperative to see how their findings translate to iPSC-derived motoneurons and whether they have any causal link to their neurodegeneration. It is expected that the impaired aminoacylation activity of FARS would lead to lower Phe-tRNA levels. Importantly, landmark recent papers in Science (PMID 34516840 , PMID 34516839) demonstrated that a deficiency in the levels of Glycyl-tRNA causes an integrative stress response that can explain (at least in part) the neurodegeneration associated with mutations in the glycyl-tRNA synthetase. In analogy, is there any stress- (or other) cell response induced by the impaired aminoacylation activity of FARS?

In the discussion, the authors are very superficial in how their findings fit the current knowledge about tRNA synthetases and neurodegeneration. They refer to the sticky mouse that displays a neurodegenerative phenotype (cerebellar Purkinje cell loss and ataxia). I am not sure their findings are in line with this model. They do not present data demonstrating that extended C4H2-mRNA-FARS interaction leads to mistranslation. No clear link with any of the other current hypotheses in the field is discussed too. It will be important that the discussion of the manuscript is substantially revised to provide a clear view of the ideas the authors have about the significance of their biochemical findings in the context of neurodegeneration and beyond.

Minor comments:

The in vitro aminoacylation assay is following the steady-state kinetics. It is expected that after a defined period of time it will reach a plateau (saturation) phase. This is not the case in the presented graph (Fig. 2E). Because the aminoacylation activity of the FARS in the presence of expanded C4G2-mRNA is impaired but not abolished, one can wonder what would happen if the incubation is for a longer time than 30 minutes. Although less likely, it is still possible that the mutants will aminoacylate all tRNA-Phe present in the mixture, but for a much longer time.

Overall, the schemes and figures presented are very well designed. Yet, some of the figures need more elaborated legends to understand their message. For example, Fig.S3 – what is illustrated in the insets and where are the selected aggregates in the nucleus?

Reviewer #3 (Remarks to the Author):

In this manuscript, Mirjana Malnar and colleagues identify a potential disruption to protein synthesis related to the C9ORF72 mutation. They show that antisense (C4G2) repeats encoded by the expanded C9ORF72 mutation bind phenylalanine-tRNA synthetase (FARS) subunit alpha (FARSA), resulting in reduced Phe charging and synthesis of Phe-rich proteins. This is a highly interesting and novel finding, with potentially relevant consequences for disease.

The experiments are well controlled and the data are clear. My main concern is that the overall significance of findings as they stand are unclear - without additional studies into the implications of FARS dysfunction, the consequences of FARSA sequestration in disease remain unknown.

Major points:

The impact of FARS sequestration by G4C2 RNA is unclear. A functional assay for any of the Phe-rich proteins displayed in Fig. 3 might be helpful in this regard. Similarly, some indication of how this

deficiency affects neuronal function would go a long ways towards establishing significance for the findings.

Can any of the identified deficiencies in protein synthesis be reversed by addition of high Phe levels or tRNA-Phe?

Several proteomics studies of C9ORF72 mutant cells, neurons and patient derived material have been performed. Even if the authors do not conduct experiments themselves, they may find evidence of disruptions to Phe-rich proteins in these datasets (for instance, PMID: 30456350, 34638725, 32059759).

The authors show convincingly that several proteins are pulled down by bait RNA consisting of 32 repeats of C4G2 RNA. It is unclear if this phenomenon is repeat length dependent; for instance, would the same sequestration occur with 8 G4C2 repeats?

Minor points:

Human genes are capitalized, while rodent genes are lowercase. All instances of C9orf72 should be changed to C9ORF72 when dealing with human cells and the human gene.

Within the introduction, the authors should clarify that the C9ORF72 mutation is the largest contributor to familial ALS and FTD, and only within Europe and Northern America.

The authors are encouraged to correlate % Phe tRNA charging from different cell types with PLA signal in each cell type in Fig 2B-D

Similarly, the authors could correlate the degree of change in expression for proteins with their Phe content in Fig 3.

In Fig 2B, please provide additional labeling within figure itself regarding the antibodies used for PLA. Also, some indicator of the cell borders or cytoplasm would be helpful, such as brightfield or a cell fill marker (GFP)

In Fig 2E, it is difficult to discriminate circles from squares. Please use dotted lines or other, more obvious indicators for different conditions.

The authors should comment on why there is not a relative increase in free Phe in Fig. 2G if it is not being used for protein synthesis in C9ORF72 mutant cells.

There are several instances in the text where the referenced figure does not correlate with the figure and/or panel. For instance, lines 156-159.

Additional replicates for WB are required in Fig. 3.

It is unclear what the two blots represent in S1C.

We are pleased to submit the revised draft of our manuscript, "Phenylalanine-tRNA aminoacylation is compromised by ALS/FTD-associated C9orf72 C4G2 repeat RNA" (manuscript ID: NCOMMS-22-40657), to Nature Communications.

Please find below our response to the comments of the reviewers.

RESPONSE TO REVIEWER 1:

In this interesting manuscript the protein interactors with long (30x) C4G2 repeats are identified. The interest of this exercise is identifying the targets that may be linked to the pathologies caused by such repeats.

Convincingly, PheRS is identified as one of such interactors, and the binding of C4G2 repeats to this enzyme is shown to modestly reduce tRNA aminoacylation. As a logical consequence of this inhibition, the authors show that the translation of Phe-rich transcripts is particularly affected because the amounts of freshly-translated Phe-rich proteins are clearly diminished with the C4G2 repeats present. This suggests that an important factor in C9orf72-related pathologies might be compromised protein synthesis machinery.

Response: We are grateful to the reviewer 1 to find our manuscript interesting and for the provided insightful comments. Please find below our response to their questions/comments/concerns.

1.1. Although the data presented is clear, my concern with the manuscript is that it does not analyze in depth what is the functional consequence of this inhibitory effect over PheRS. In the discussion, the authors refer to other examples where compromised ARS function has an impact on fidelity, but the possibility of mistranslation at Phe codons caused by C4G2 repeats is not explored. Similarly, the fascinating sensitivity of neuronal tissues to translation defects is mentioned, but whether neuronal cells are more sensitive than other cells to C4G2 repeats is not analyzed.

Response to 1.1.: We agree with this comment. All three reviewers had similar comments regarding the functional/disease consequences of our findings and suggested a number of directions to explore.

Therefore, we added the following experiments:

- 1) FARSA loss of function knockdown experiments on HEK293 cells.
- 2) FARSA loss of function knockdown experiments on differentiated motor neuron-like cell line NSC-34.
- 3) Analysis of Phe-rich proteins in disease relevant cerebellum samples from C9orf72 mutation positive ALS/FTD patients in comparison to C9orf72 negative ALS/FTD patients.
- 4) Analysis of the cell stress pathway and mislocalisation of TDP-43.
- 5) Analysis of length dependence of FARSA-C4G2 interaction.
- 6) Analysis of Phe expression from proteomic experiment of C9orf72-patient derived lymphoblastoid cell lines showing increasing percentage of downregulated proteins with increasing percentage of Phe in proteins.

We performed partial knockdown experiments both on HEK293 cells and differentiated motor neuron-like cell line NSC-34, discussed more in detail in question 1.2. We observed reduced

expression levels of Phe-rich proteins, the same as in C9orf72-patient derived lymphoblastoid cell lines. The observed expression level of proteins is affected by many other factors (effects on transcription, RNA processing, other aspects of protein translation, protein turnover, etc.), which may have greater effect on the protein level than the reduction of tRNA^{Phe} charging but with knockdown experiment we show that FARSA dysfunction can lead to observed effects. The expanded RNA repeats are not present only in the neurons but also in other tissues of C9orf72 ALS/FTD patients, however, with great variation in their length and so far, not a clear correlation between presence of these repeats in various tissues and disease onset or progression¹⁻⁶. Both sense and antisense RNA foci are present in the disease; however, they differ in the distribution also among different neuron types⁷⁻⁹. Additionally, ALS and FTD are both systemic diseases which means that some irregularities are non-cell autonomous and other cell types could also affect neurons¹⁰. Studies so far have not found the reason for neither expanded repeats being more toxic to neuronal cells compared to others nor for greater sensitivity of these cells to impairments of ARSs compared to other cell types.

Most importantly we showed potential connection of FARSA dysfunction to C9orf72 positive ALS/FTD cases. We investigated the expression levels of Phe-rich proteins in cerebellum of C9orf72-mutation positive ALS/FTD patients in comparison to C9orf72 negative ALS/FTD (Figure 4B, Figure S8C). We observed a significant decrease of Phe-rich proteins TSPAN5 and ALG10B, which importantly supports disease relevance of our findings. The antibodies for PXMP2 and GOLT1B didn't work well on post mortem tissue to allow their quantification, which we also mention in the manuscript.

We hope reviewer 1 will appreciate additional functional experiments that we have done. In this manuscript we focused on the interaction of C4G2 repeats and FARSA subunit of FARS protein. FARSA subunit is responsible for attachment of Phe to its cognate tRNA, whereas FARSB subunit has an editing function in preventing misacylation¹¹. We appreciate that we did not stress this point sufficiently in the manuscript, we have now added this information already in the introduction and hopefully this made the revised version clearer. The study focused on reduction in Phe-tRNA aminoacylation as opposed to mistranslational effects. This was due to observed interaction of C4G2 RNA repeats with FARSA subunit in multiple different experiments, but not with FARSB subunit, which would carry out the editing effect in case of mistranslation. The hypothesis was therefore, that the amino acid attachment to tRNA^{Phe} would be overall reduced. As you can appreciate, the function of FARSA in the presence of C4G2 RNA repeats was analysed in depth. This was done both in an isolated system (tRNA^{Phe} aminoacylation assay) and in C9orf72 patient-derived cells on two levels – tRNA^{Phe} aminoacylation and consequent incorporation of Phe into cell proteome.

1.2. Finally, the relative importance of PheRS inhibition with respect to the rest of C4G2 interactions is not measured, although it would be relatively simple to silence PheRS expression and look for the emergence of disease-related phenotypes.

Response to 1.2.: We have taken this insightful comment into consideration. We performed experiments on two different knockdown cell lines. With partial FARSA knockdown experiments in HEK293 cells (Figure 3B, Figure S7) we now show that FARSA loss of function is comparable to effect observed in patient cell lines (Figure 3C, Figure S8A) and tissues (Figure 4B, Figure S8C) in regard to expression levels of Phe-rich proteins. To evaluate whether the

same effect of FARSA dysfunction is also seen on neuronal cell lines we have performed knockdown experiment on differentiated motor neuron-like cell line NSC-34 (Figure 4A, Figure S9). With lower knockdown efficiency compared to HEK293 cells (around 20 % in differentiated NSC-34 versus 50% in HEK293) we observed a significant reduction in PXMP2, GOL1B and ALG10B Phe-rich proteins but not controls.

RESPONSE TO REVIEWER 2:

The manuscript of Malnar et al describes new aspects of the toxicity caused by the C4G2-repeat expansion in the C9orf72 gene causing AFD/ALS phenotype. The authors focus on studying the cytoplasmic interactions of the C4G2 antisense repeat RNA, a disease aspect that is underresearched, and discover that one of the binding partners is the phenylalanyl-tRNA synthetase (FARS). The neomorphic interaction is detected in C9orf72-derived fibroblasts, lymphoblasts, and iPSCs and is thus a ubiquitous finding. This interaction is differential, as it is not present when probing against other ARS, such as aspartyl- leucyl- and glutamyl-prolyl-tRNA synthetases. Further detailed biochemical studies in vitro and in different cellular model systems reveal a direct interaction between the alpha-subunit of FARS and the extended C4G2-transcript that has functional consequences on the FARS holoenzyme. The aminoacylation activity of FARS is significantly reduced both in in vitro aminoacylation assays and in total lysates from lymphoblasts of C9orf72-patients. Furthermore and importantly, the impaired aminoacylation activity of FARS has functional consequences on the global proteome of both HEK293 cell models expressing extended C4G2 repeat -mRNA and C9orf72 lymphoblasts. The authors demonstrate both in HEK293 cell models and C9orf72-lymphoblasts underrepresentation of Phe-rich proteins in the proteome (~20% reduction in their expression) at the background of normal phenylalanine amino acid cellular levels. Their hypothesis is that these proteome changes might have negative implications for the development of the C9orf72-related pathologies.

From a biochemical point of view, this is a very solid study that makes an original contribution to the field of both protein synthesis and neurodegeneration. The findings are original and somehow unexpected, but the provided evidence makes them well justified.

Response: We thank the reviewer 2 to find our study important for the field and for the provided suggestions. We have taken the useful comments into account and revised our manuscript accordingly. Please find the answer to each of your questions below.

2.1 From a mechanistic perspective, the impact of the findings – neither for general cell biology nor for neuronal biology - is unclear as no experimental data are provided nor hypotheses are raised. The findings pose several questions that await their answer before we can judge their contribution to enlightening the C9orf72-related neurodegeneration. It would be important that the authors provide experimental data or at least discuss their view on how their findings would be causally related to C9orf72 brain pathology in the patients. Some of the appealing open questions are: Why do neurons only suffer from the decrease of Phe-rich proteins while this phenomenon was detected in non-neuronal cell populations of the patients too?

Response to 2.1.: We appreciate the reviewer's insightful comment. The functional consequences are important aspect of our research. We therefore conducted additional FARS loss of function knockdown experiments on two different cell types (Figure 3B, Figure 4A) discussed more in response 2.2. and Phe-rich protein expression studies in human post-mortem cerebellum tissues from C9orf72 mutation positive ALS/FTD patients (Figure 4B, Figure S8C) to check for the relevance of our results in C9orf72 pathology. We observed the reduction of Phe-rich proteins in knockdown experiments. Importantly when we checked for the expression of Phe-rich proteins in cerebellum we observed a reduction for TSPAN5 and ALG10B in C9orf72 mutation positive ALS and FTD patients (Figure 4B, Figure S8C). The antibodies for other two proteins didn't work well on the post-mortem tissue which is stated in the manuscript. Additionally, we performed proteomics study on C9orf72-mutation positive lymphoblastoid cell line to support previously obtained results. We observed that the ratio of downregulated to upregulated genes increases with the percentage of Phe in proteins.

The observed expression level of proteins is affected by many other factors (effects on transcription, RNA processing, other aspects of protein translation, protein turnover, etc.), which may have greater effect on the protein level than the reduction of tRNA^{Phe} charging. But we show that FARS1 dysfunction has an important role. The expanded RNA repeats are not present only in the neurons but also in other tissues of C9orf72 ALS/FTD patients, however, with great variation in their length and so far, not a clear correlation between presence of these repeats in various tissues and disease onset or progression¹⁻⁶. Both sense and antisense RNA foci are present in the disease; however, they differ in the distribution also among different neuron types⁷⁻⁹. Additionally, ALS and FTD are both systemic diseases which means that some irregularities are non-cell autonomous and other cell types could also affect neurons¹⁰. Studies so far have not found the reason for neither expanded repeats being more toxic to neuronal cells compared to others nor for greater sensitivity of these cells to impairments of ARSs compared to other cell types.

We have included new results and improved our discussion in response to this question.

2.2. Is there a causal relationship between the partial loss of FARS1 activity and the C9orf72-neurodegeneration? The authors provide experiments in iPSC cells from patients, but it would be imperative to see how their findings translate to iPSC-derived motoneurons and whether they have any causal link to their neurodegeneration. It is expected that the impaired aminoacylation activity of FARS1 would lead to lower Phe-tRNA levels. Importantly, landmark recent papers in Science (PMID 34516840, PMID 34516839) demonstrated that a deficiency in the levels of Glycyl-tRNA causes an integrative stress response that can explain (at least in part) the neurodegeneration associated with mutations in the glycyl-tRNA synthetase. In analogy, is there any stress- (or other) cell response induced by the impaired aminoacylation activity of FARS1?

Response to 2.2.: We appreciate this comment. It is important to place our findings into the context of disease, beyond patient derived lymphoblastoid cell lines, though we think that lymphoblastoid cell lines have proved themselves as an important tool in our study. We therefore evaluated disease relevance in human post-mortem cerebellum tissue in C9orf72 mutation positive ALS/FTD patients and showed the effect observed on cell lines also in patient tissue (Figure 4B, Figure S8C).

To observe if partial loss of FARS1 function is related to these results we performed knockdown experiments on HEK293 and observed a significant reduction of four Phe-rich proteins (Figure 3B, Figure S7).

In order to observe effects in motor neuron – like cells we evaluated the effect of FARS1 loss of function in differentiated NSC-34 cells (Figure 4A, Figure S9). In a partial knockdown of 74 ± 5.9 % FARS1 expression (Figure S9A) we observed reduction in expression of all tested Phe-rich proteins. Apart from TSPAN5, tested proteins showed significant decrease in expression in three biological replicates. To analyse the late aspects of C9orf72 effect we focused more on neuropathology and observed reduction of Phe-rich protein expression as well.

In accordance with the second part of the comment we have examined the integrated stress response. With qPCR analysis of factors *ASNS*, *GPT2* and *eIF4EBP1* (Figure S10) which transcription increases in amino acid stress and unfolded protein response, acting downstream of eIF2 α kinase and ATF4¹², we checked if there is activation of integrated stress response in C9orf72-patient derived lymphoblastoid cell lines, HEK293 and NSC-34 knockdown cells. We didn't observe increased transcription levels in any experiment which indicates that our mechanism affects cells below stress level. In HEK293 KD cells we also checked for formation

of stress granule and we did not observe their formation with IF staining for stress granule marker PABPC1. Since we didn't get any positive data supporting the presence of stress responses, we didn't evaluate this further. These results have been added to the manuscript and we improved our discussion regarding the integrated stress response.

2.3. In the discussion, the authors are very superficial in how their findings fit the current knowledge about tRNA synthetases and neurodegeneration. They refer to the sticky mouse that displays a neurodegenerative phenotype (cerebellar Purkinje cell loss and ataxia). I am not sure their findings are in line with this model. They do not present data demonstrating that extended C4H2-mRNA-FARS interaction leads to mistranslation. No clear link with any of the other current hypotheses in the field is discussed too. It will be important that the discussion of the manuscript is substantially revised to provide a clear view of the ideas the authors have about the significance of their biochemical findings in the context of neurodegeneration and beyond.

Response to 2.3.: We appreciate this comment and have taken it into consideration. As there are only a few studies on FARS protein in human diseases, the correlation of some aspects was indeed done based on the knowledge on other aminoacyl-tRNA synthetases, as many of them have been associated with neurodegeneration^{13,14}. Moreover, this study was focused on function of FARSA subunit, impairment of which would result in overall reduced tRNA^{Phe} aminoacylation, as opposed to mistranslation, which would most likely be a consequence of impairment in editing subunit of FARS – FARSB¹¹. Other mechanisms could also affect mistranslation like changes in the wobble position, misacylation on other tRNAs and problems in ribosome quality control. Mistranslations are a broad field that we have not yet focused on. We excluded the part about misacylation aspect and improved our discussion according to the remarks. We added our observations on how reduction in Phe-rich proteins, each with their own functions, could affect neurodegenerative pathways.

2.4. The in vitro aminoacylation assay is following the steady-state kinetics. It is expected that after a defined period of time it will reach a plateau (saturation) phase. This is not the case in the presented graph (Fig. 2E). Because the aminoacylation activity of the FARS in the presence of expanded C4G2-mRNA is impaired but not abolished, one can wonder what would happen if the incubation is for a longer time than 30 minutes. Although less likely, it is still possible that the mutants will aminoacylate all tRNA-Phe present in the mixture, but for a much longer time.

Response to 2.4.: Thank you for your valuable insight. We have now extended the in vitro aminoacylation assay to 90 minutes, with measurements taken at 10, 20, 30, 60 and 90 min. The data are presented in the Figure 2E. With the additional time points it is now shown that saturation is reached between 60 and 90 min for all the samples. In the presence of 32×C4G2 RNA repeats the levels of aminoacylated tRNA^{Phe} in the sample was still lower than in control, showing the inhibition of FARS tRNA-aminoacylation function even after prolonged incubation. The enzyme cannot make up entirely for the lower efficiency caused by the inhibition by getting more time to charge all the tRNAs in the system. In these longer periods some of the enzyme or tRNA may get damaged or otherwise decommissioned which might result in lower plateau. The new in vitro experiment suggested by the reviewer further solidifies our hypothesis that presence of C4G2 expanded hexanucleotide RNA repeats inhibits the tRNA aminoacylation function of FARS protein and could impact the protein synthesis also in the long run.

2.5. Overall, the schemes and figures presented are very well designed. Yet, some of the figures need more elaborated legends to understand their message. For example, Fig.S3 – what is illustrated in the insets and where are the selected aggregates in the nucleus?

Response to 2.5.: We added labelling and description to Figure S3. We hope you will find the figures and legends better presented in the revised manuscript.

RESPONSE TO REVIEWER 3:

In this manuscript, Mirjana Malnar and colleagues identify a potential disruption to protein synthesis related to the C9ORF72 mutation. They show that antisense (C4G2) repeats encoded by the expanded C9ORF72 mutation bind phenylalanine-tRNA synthetase (FARS) subunit alpha (FARSA), resulting in reduced Phe charging and synthesis of Phe-rich proteins. This is a highly interesting and novel finding, with potentially relevant consequences for disease.

The experiments are well controlled and the data are clear. My main concern is that the overall significance of findings as they stand are unclear - without additional studies into the implications of FARS dysfunction, the consequences of FARSA sequestration in disease remain unknown.

Response: We are grateful to the reviewer for the insightful comments provided. Please find the answers to them below.

3.1. The impact of FARS sequestration by G4C2 RNA is unclear. A functional assay for any of the Phe-rich proteins displayed in Fig. 3 might be helpful in this regard. Similarly, some indication of how this deficiency affects neuronal function would go a long way towards establishing significance for the findings.

Response to 3.1: We appreciate this comment. We have done additional experiments to try to answer the question of impact from mechanistic to disease association. In order to show that the observation is analogous to partial loss of function of FARSA, we have performed FARSA knockdown experiments on HEK293 cells (Figure 3B, Figure S7) where we observed a significant reduction in all selected Phe-rich proteins- similar results as in C9orf72- patient derived lymphoblastoid cell lines. To evaluate whether same effect of FARSA dysfunction is also seen on neuronal cell lines we have performed knockdown experiment on differentiated motor neuron-like cell line NSC-34 (Figure 4A, Figure S9). With lower knockdown efficiency compared to HEK293 cells (around 20 % in differentiated NSC-34 versus 50% in HEK293) we observed a significant reduction in three Phe-rich proteins. These results show that partial FARSA dysfunction affect different cell types. ALS and FTD are both systemic diseases which means that irregularities in other types of cells could be observed. For example, sense and antisense RNA foci expression was found both in neuronal cells as well as supporting cells where repeats are differently expressed and differ in length^{15,16}.

To further observe potential connection of FARSA dysfunction to the disease we investigated the expression levels of Phe-rich proteins in cerebellum of patients with C9orf72 mutation positive ALS/FTD in comparison to C9orf72 negative ALS/FTD (Figure 4B, Figure S8C). We observed a significant decrease in two out of four Phe-rich proteins which shows disease relevance of our findings. The other two antibodies didn't work well on post-mortem tissue which is stated in the manuscript. Post-mortem tissue is a complex system compared to our

clear association made previously in transfected HEK293T cells, where expression of C4G2 repeats is the only difference between control and C9orf72 model.

We added our observations on how reduction in Phe-rich proteins, each with their own functions, could affect neurodegenerative pathways in the revised discussion.

3.2. Can any of the identified deficiencies in protein synthesis be reversed by addition of high Phe levels or tRNA-Phe?

Response to 3.2: Thank you for your comment. We tried to address this by treating C9orf72 and control lymphoblastoid cell lines with excessive amount of Phe (data not shown in the manuscript). This did not result in restoring the levels of PXMP2 protein which was the Phe-rich protein with the biggest change in expression. This was tested for 30x and 100x excessive amount of phenylalanine. Since we did not observe an effect in this protein and the protein was still reduced, we didn't continue with the analysis of others due to time constraint. Also, in click-experiment presented in the article we added additional amount of azidophenylalanine which still resulted in decreased expression of Phe-containing proteins. Transfection/transduction of lymphoblastoid cell lines was very inefficient in our hands making the tRNA^{Phe} overexpression experiment difficult to carry out.

3.3. Several proteomics studies of C9ORF72 mutant cells, neurons and patient derived material have been performed. Even if the authors do not conduct experiments themselves, they may find evidence of disruptions to Phe-rich proteins in these datasets (for instance, PMID: 30456350, 34638725, 32059759).

Response to 3.3: We are grateful for the suggestion. We have checked the suggested literature and PMID: 34638725 (fibroblasts) was found to be useful for the suggested analysis, however we did not find expression of our selected Phe-rich proteins. Of the other two suggested datasets, PMID: 30456350 is an interactomic study of polyGR and PR and PMID: 32059759 is based on sense repeat cell model.

Therefore, we also carried out MS analysis on six C9orf72- patient derived lymphoblastoid cell lines and six controls which we used in our experiments before. For our analysis we used all proteins exhibiting log₂Fc over 0.15 which correlates to around 10% change in expression. We observed that the ratio of downregulated compared to upregulated genes increases with the increase in % Phe in proteins (Figure 4C). We hope that with this additional study we sufficiently answer the question.

3.4. The authors show convincingly that several proteins are pulled down by bait RNA consisting of 32 repeats of C4G2 RNA. It is unclear if this phenomenon is repeat length dependent; for instance, would the same sequestration occur with 8 G4C2 repeats?

Response to 3.4: Thank you for your comment. We intentionally focused on the long RNA repeats, as these are more relevant to the disease which gives more strength to our study. In order to address this question, we performed RNA-pull down assay on HEK293 cell lysate with 8x, 24x and 32x C4G2 repeats. Since we focused on the interaction between the antisense repeats and FARSA protein we checked for the binding of FARSA on short repeats. We observed strong binding of FARSA protein on 32x and 24x repeats but barely detectable for 8x repeats. Previous studies have been done with 4 C4G2 RNA repeats^{7,8,17}. The exact number of repeats causing the disease has not yet been defined. The average number of repeats in healthy individuals is 5-10 and reaches up to 25². In the previous studies 24-30 repeat units

have been proposed to present a risk factor for sALS, PD, and essential tremor (ET). Moreover, there are reports of ALS cases carrying 24 and 28 repeats, as well as FTD cases with 20-22 repeats. Even though the reported numbers of repeats are greatly variable, it is estimated that most expansions in disease are several hundred to several thousand repeats long⁶.

Minor points:

3.5. Human genes are capitalized, while rodent genes are lowercase. All instances of C9orf72 should be changed to C9ORF72 when dealing with human cells and the human gene.

Response to 3.5: Thank you for your comment. In the case of C9orf72 there is an exception: "Symbols of human genes must only contain uppercase Latin letters and Arabic numerals with sole exception of C#orfs¹⁸."

3.6 Within the introduction, the authors should clarify that the C9ORF72 mutation is the largest contributor to familial ALS and FTD, and only within Europe and Northern America.

Response to 3.6: Thank you for pointing out the missing information. We have now added this information to the introduction.

3.7. The authors are encouraged to correlate % Phe tRNA charging from different cell types with PLA signal in each cell type in Fig 2B-D

Response to 3.7: Thank you for your suggestion. However, the tRNA^{Phe} charging level changes were calculated only in the lymphoblastoid cell lines. Moreover, the PLA signal ratio between C9orf72 and control cells as well as tRNA^{Phe} charging level changes are calculated as means over multiple C9orf72 and control lymphoblastoid cell lines as opposed to particular combinations of them, which would be needed for calculating correlation coefficient. In this case the correlation coefficient would heavily depend on chosen combinations of C9orf72 and control lymphoblastoid cell lines to be directly compared. Additionally, the nature of PLA is that it is semiquantitative and useful only in direct comparison experiments.

3.8. Similarly, the authors could correlate the degree of change in expression for proteins with their Phe content in Fig 3.

Response to 3.8: Thank you for your suggestion. We weren't able to correlate the degree of change in expression for proteins with their Phe content. We didn't see any pattern so we decided not to include this analysis. There are additional factors that play a role in expression of these proteins and because of that the lack of correlation in the number of proteins we tested is something that is not surprising.

3.9. In Fig 2B, please provide additional labeling within figure itself regarding the antibodies used for PLA. Also, some indicator of the cell borders or cytoplasm would be helpful, such as brightfield or a cell fill marker (GFP).

Response to 3.9: Thank you for your comment. We added the antibody used/protein detected for each part of the figure above the graphs. We hope this will improve the clarity of the figure. For the sake of figure visibility and clarity, we kept the original figure (Figure 2 B-D) in the article. However, we are adding the figure with included brightfield images as part of this

response. We sincerely hope that is acceptable/sufficient (Response Figure 1, also labelled as B-D for clarity).

3.10. In Fig 2E, it is difficult to discriminate circles from squares. Please use dotted lines or other, more obvious indicators for different conditions.

Response to 3.10: Thank you for your suggestion. We have changed the labelling in Figure 2E. We changed the labelling of each condition with different shape: NoRNA -triangles; RFP RNA- circles; C4G2 RNA- squares. For faster differentiation between the conditions, we also changed the colour scheme: NoRNA – black and gray; RFP RNA- shades of green; C4G2 RNA- shades of blue. We have also marked different concentrations in different shades of colour from the highest being the darkest to lowest being the lightest. We hope this increases the clarity of the graph.

3.11. The authors should comment on why there is not a relative increase in free Phe in Fig. 2G if it is not being used for protein synthesis in C9ORF72 mutant cells.

Response to 3.11: Thank you for your comment. There are multiple factors contributing to amino acid homeostasis in mammalian cells including import, export, metabolism, synthesis of nonessential aminoacids, protein synthesis and protein breakdown. These processes depend on regulatory mechanisms responding to amino acids or their metabolites, therefore establishing homeostasis in the cells. Concentrations of amino acids, like other metabolites, are kept within narrow limits and deviations in these concentrations usually occur in rare disorders¹⁹. Therefore, decreased attachment of Phe to its cognate tRNA, and potential rise of its concentration in cell would probably be compensated by lesser intake of Phe by the cell, increased metabolism of the Phe or other regulatory mechanisms that keep Phe in certain concentration limits, unless regulation mechanisms of the cell would also be impaired. In the discussion we have added the following: “The level of free Phe available in cells was not raised as it is most likely compensated by the cell homeostasis mechanisms¹⁹”.

3.12. There are several instances in the text where the referenced figure does not correlate with the figure and/or panel. For instance, lines 156-159.

Response to 3.12: Thank you for your comment. We apologize for the mistake, and have now corrected the references in the text.

3.13. Additional replicates for WB are required in Fig. 3.

Response to 3.13: Thank you for your comment. Replicates of the WB are presented in the supplemental material Figure S8A. In this experiment each lymphoblastoid sample presents one biological control since they were obtained from different C9orf72-mutation positive patients or controls. In order to avoid impact of potential technical errors we performed mixed model analysis on two technical repeats. The technical repeats are not independent since they derive from the same cell lines and are not treated. We showed that all Phe-rich proteins are significantly decreased after this analysis that eliminates the technical errors. We believe this is enough replicates for this analysis.

3.14. It is unclear what the two blots represent in S1C.

Response to 3.14: Thank you for your comment. The two blots represent two replicates of the RNA pull-down assay performed on the human brain lysate. We added the following sentence in the figure legend: “For each protein two side by side western blot membranes

are presented from two replicates of RNA pull-down assay.” We hope that makes the figure clearer.

1. DeJesus-Hernandez, M. *et al.* Expanded GGGGCC hexanucleotide repeat in noncoding region of C9ORF72 causes chromosome 9p-linked FTD and ALS. *Neuron* **72**, 245–256 (2011).
2. Gao, F., Almeida, S. & Lopez-Gonzalez, R. Dysregulated molecular pathways in amyotrophic lateral sclerosis–frontotemporal dementia spectrum disorder. *EMBO J.* **36**, 2931 (2017).
3. Gitler, A. D. & Tsuiji, H. There has been an awakening: Emerging mechanisms of C9orf72 mutations in FTD/ALS. *Brain Res.* **1647**, 19 (2016).
4. Ranganathan, R. *et al.* Multifaceted Genes in Amyotrophic Lateral Sclerosis-Frontotemporal Dementia. *Front. Neurosci.* **14**, (2020).
5. Renton, A. E. *et al.* A hexanucleotide repeat expansion in C9ORF72 is the cause of chromosome 9p21-linked ALS-FTD. *Neuron* **72**, 257–268 (2011).
6. Van Mossevelde, S., van der Zee, J., Cruts, M. & Van Broeckhoven, C. Relationship between C9orf72 repeat size and clinical phenotype. *Curr. Opin. Genet. Dev.* **44**, 117–124 (2017).
7. Cooper-Knock, J. *et al.* Sequestration of multiple RNA recognition motif-containing proteins by C9orf72 repeat expansions. *Brain* **137**, 2040–2051 (2014).
8. Cooper-Knock, J. *et al.* Antisense RNA foci in the motor neurons of C9ORF72-ALS patients are associated with TDP-43 proteinopathy. *Acta Neuropathol.* **130**, 63–75 (2015).
9. Zu, T. *et al.* RAN proteins and RNA foci from antisense transcripts in C9ORF72 ALS and frontotemporal dementia. *Proc. Natl. Acad. Sci. U. S. A.* **110**, (2013).
10. Van Harten, A. C. M., Phatnani, H. & Przedborski, S. Non-cell-autonomous pathogenic mechanisms in amyotrophic lateral sclerosis. *Trends Neurosci.* **44**, 658–668 (2021).
11. Guo, M. & Schimmel, P. Structural analyses clarify the complex control of mistranslation by

- tRNA synthetases. *Curr. Opin. Struct. Biol.* **22**, 119–126 (2012).
12. Waldron, A. L., Wilcox, C. E., Francklyn, C. S. & Ebert, A. M. Knock-down of Histidyl-tRNA Synthetase causes cell cycle arrest and apoptosis of neuronal progenitor cells in vivo. *Front. Cell Dev. Biol.* **7**, 67 (2019).
 13. Lee, J. W. *et al.* Editing-defective tRNA synthetase causes protein misfolding and neurodegeneration. *Nat.* *2006 4437107* **443**, 50–55 (2006).
 14. Van Dijk, K. D. *et al.* The Proteome of the Locus Ceruleus in Parkinson's Disease: Relevance to Pathogenesis. *Brain Pathol.* **22**, 485–498 (2012).
 15. Gendron, T. F. & Petrucelli, L. Disease Mechanisms of C9ORF72 Repeat Expansions. *Cold Spring Harb. Perspect. Med.* 1–21 (2018).
 16. McEachin, Z. T., Parameswaran, J., Raj, N., Bassell, G. J. & Jiang, J. RNA-mediated toxicity in C9orf72 ALS and FTD. *Neurobiol. Dis.* **145**, 105055 (2020).
 17. Haeusler, A. R. *et al.* C9orf72 nucleotide repeat structures initiate molecular cascades of disease. *Nat.* *2014 5077491* **507**, 195–200 (2014).
 18. Bruford, E. A. *et al.* Guidelines for Human Gene Nomenclature. *Nat. Genet.* **52**, 754 (2020).
 19. Bröer, S. & Gauthier-Coles, G. Amino Acid Homeostasis in Mammalian Cells with a Focus on Amino Acid Transport. *J. Nutr.* **152**, 16–28 (2022).

Response Figure 1 FARSA interacts with antisense RNA in the nucleus and cytoplasm of C9orf72 patient-derived cells and the presence of C4G2 RNA reduces aminoacylation of tRNA^{Phe}. The RNA-protein proximity ligation assay (PLA) shows increased PLA signals (magenta) in C9orf72 patient-derived cells with expanded repeats relative to controls: 0.78 ± 0.36 in control versus 2.49 ± 0.92 in C9orf72 patient-derived fibroblasts (B), 4.85 ± 0.77 in control versus 44.98 ± 3.95 in C9orf72 patient-derived lymphoblastoid cell lines (C), and 0.31 ± 0.08 in control versus 6.00 ± 1.12 in C9orf72 patient-derived induced pluripotent stem cells (iPSCs) (D). For all three cell types, C9orf72 patient-derived and control cells are displayed above and below, respectively.

REVIEWERS' COMMENTS

Reviewer #1 (Remarks to the Author):

The authors have fully addressed my concerns, and the manuscript is now more complete and convincing.

Reviewer #2 (Remarks to the Author):

The authors addressed thoroughly the raised comments and added a good amount of novel data. The new experiments using patient-derived tissue add significant weight to their findings.

My only minor comment is to replace "cerebral ataxia" on line 91 in the Introduction with "cerebellar ataxia".

Reviewer #3 (Remarks to the Author):

The authors have addressed many of the concerns that were raised on the initial round of reviews, and the revised manuscript is stronger. My only remaining concern is with the new proteomics data shown in Fig. 4B -- although this is an appropriate method, it is strange that the authors were unable to confirm downregulation of specific Phe-containing proteins, including those illustrated in previous figures (i.e. Fig. 3c). The ratio of downregulated/upregulated proteins with particular %Phe content is interesting, but may require additional explanation for readers to grasp the implications.

Point-by-point response to the reviewers:

RESPONSE TO REVIEWER 1:

The authors have fully addressed my concerns, and the manuscript is now more complete and convincing.

Response: We thank the reviewer for the positive comment of our additional work.

RESPONSE TO REVIEWER 2:

The authors addressed thoroughly the raised comments and added a good amount of novel data. The new experiments using patient-derived tissue add significant weight to their findings.

My only minor comment is to replace "cerebral ataxia" on line 91 in the Introduction with "cerebellar ataxia".

Response: We thank the reviewer for the comment. We have corrected the expression on line 91.

RESPONSE TO REVIEWER 3:

The authors have addressed many of the concerns that were raised on the initial round of reviews, and the revised manuscript is stronger. My only remaining concern is with the new proteomics data shown in Fig. 4B -- although this is an appropriate method, it is strange that the authors were unable to confirm downregulation of specific Phe-containing proteins, including those illustrated in previous figures (i.e. Fig. 3c). The ratio of downregulated/upregulated proteins with particular %Phe content is interesting, but may require additional explanation for readers to grasp the implications.

Response: We thank the reviewer for this comment. Out of the four examined proteins only two were detected with MS analysis: PXMP2 and GOLT1B. For these proteins downregulation was observed but not significant. This could be due to low peptide abundance. For both we

only detected 2 peptides with low abundances and one of the peptides is the same as the other but with 1 missed cleavage (see below Figure 1a, b). The other two proteins: TSPAN5 and ALG10B were not detected with our analysis. This suggests that PXMP2 and GOLT1B are not abundant enough to be detected by the current LC-MS methods in sufficient amounts to get reliable results. In complex cell lysates WB is more sensitive than mass spectrometry especially for less abundant proteins. Since most of Phe rich proteins were not highly detected with our MS analysis, we have decided to check if the trend of increasing Phe affects the expression level of proteins on larger sample of proteins. And we observed that the number of downregulated proteins increases with the increasing percentage of Phe.

a)

Sequence	# PSMs	Master Pr:	Positions in Master	Master Protein Descriptions	# Missed Cleavages	Abundances (Normalized)											
						F1_Sample	F2_Sample	F3_Sample	F4_Sample	F5_Sample	F6_Sample	F7_Sample	F8_Sample	F9_Sample	F10_Sample	F11_Sample	F12_Sample
RVPVLGSLNLPGIR	12	Q9Y3E0	Q9Y3E0 [111-125]	Vesicle transport protein GOT1B OS=Homo sapiens	1	1.185e5	1.729e5	2.292e5	1.213e5	1.640e5	1.835e5	7.831e5	2.057e5	2.181e5	2.135e5	2.280e5	1.838e5
VPVLGSLNLPGIR	7	Q9Y3E0	Q9Y3E0 [112-125]	Vesicle transport protein GOT1B OS=Homo sapiens	0	5.711e5	6.414e5	1.146e5	8.364e5	9.412e5		9.498e5	7.114e5	9.514e5	7.705e5	1.549e5	6.743e5

b)

Sequence	# PSMs	Master Pr:	Positions in Master	Master Protein Descriptions	# Missed Cleavages	Abundances (Normalized)											
						F1_Sample	F2_Sample	F3_Sample	F4_Sample	F5_Sample	F6_Sample	F7_Sample	F8_Sample	F9_Sample	F10_Sample	F11_Sample	F12_Sample
ALAQYLLFLR	1	Q9NR77	Q9NR77 [21-30]	Peroxisomal membrane protein 2 OS=Homo sapiens	0	2.582e5	2.991e5	1.766e5		2.276e5			4.041e5	7.404e5		3.426e5	
AEAGLGALPR	5	Q9NR77	Q9NR77 [10-19]	Peroxisomal membrane protein 2 OS=Homo sapiens	0	7.851e5	5.795e5	5.339e5	6.457e5	6.640e5	6.488e5	8.210e5	6.005e5	6.327e5	6.226e5	5.694e5	1.035e5

Figure 1: Results from Mass spectrometry for a) GOLT1B and b) PXMP2. F1-F6 are samples from C9orf72- derived lymphoblastoid cell lines and F7-F12 from controls.